# Sustainable production of benzene from lignin

Qinglei Meng[1 ✉], Jiang Yan[1,2], Ruizhi Wu[1,2], Huizhen Liu [1,2], Yang Sun[3], NingNing Wu[3], Junfeng Xiang [3], Lirong Zheng[4], Jing Zhang[4] & Buxing Han [1,2,5 ✉]

Benzene is a widely used commodity chemical, which is currently produced from fossil resources. Lignin, a waste from lignocellulosic biomass industry, is the most abundant renewable source of benzene ring in nature. Efficient production of benzene from lignin, which requires total transformation of $C_{sp2}$-$C_{sp3}$/$C_{sp2}$-O into C-H bonds without side hydrogenation, is of great importance, but has not been realized. Here, we report that high-silica HY zeolite supported RuW alloy catalyst enables in situ refining of lignin, exclusively to benzene via coupling Bronsted acid catalyzed transformation of the $C_{sp2}$-$C_{sp3}$ bonds on the local structure of lignin molecule and RuW catalyzed hydrogenolysis of the $C_{sp2}$-O bonds using the locally abstracted hydrogen from lignin molecule, affording a benzene yield of 18.8% on lignin weight basis in water system. The reaction mechanism is elucidated in detail by combination of control experiments and density functional theory calculations. The high-performance protocol can be readily scaled up to produce 8.5 g of benzene product from 50.0 g lignin without any saturation byproducts. This work opens the way to produce benzene using lignin as the feedstock efficiently.

[1] Beijing National Laboratory for Molecular Sciences, CAS Laboratory of Colloid and Interface and Thermodynamics, CAS Research/Education Center for Excellence in Molecular Sciences, Institute of Chemistry, Chinese Academy of Sciences, Beijing, China. [2] School of Chemical Science, University of Chinese Academy of Sciences, Beijing, China. [3] Center for Physicochemical Analysis and Measurement, Chinese Academy of Sciences, Beijing, China. [4] Institute of High Energy Physics, Chinese Academy of Sciences, Beijing, China. [5] Shanghai Key Laboratory of Green Chemistry and Chemical Processes, School of Chemistry and Molecular Engineering, East China Normal University, Shanghai, China. ✉email: mengqinglei@iccas.ac.cn; hanbx@iccas.ac.cn

Nowadays, benzene is an indispensable commodity in chemical industry with world production of more than 61 million metric tons in 2019[1] (www.statista.com/statistics/1108114/global-benzene-capacity). The global benzene demand is anticipated to grow with a rate of 2.9% annually in the next decade[1] (www.chemanalyst.com/industry-report/benzene-market-56). Especially in the manufacturing industry, benzene is widely used in many sectors, where it is combined and processed with other basic chemicals (e.g. ethylene and propylene) to produce valuable consumer goods, such as clothing, packaging, car parts, building materials, pharmaceuticals, cosmetics, flame retardant, compact discs, eyeglass lenses, carpet, medical implants, foam insulation, adhesives, footwear, contact lens, dyes, agrochemicals (Supplementary Fig. 1)[1,2] (www.statista.com/statistics/1108114/global-benzene-capacity), (www.chemanalyst.com/industry-report/benzene-market-56). Currently, benzene is dominantly produced from petroleum and coal via catalytic reforming, steam cracking and toluene disproportionation processes, as well as coal processing (Supplementary Note 1) (Fig. 1a)[3–5]. Besides, there is also ongoing research to convert methane to benzene (Fig. 1b)[6–8]. However, all of the above routes depend on fossil resource, and has several disadvantages, such as complicated and severe conditions, high energy consumption and serious environmental pollution[9–11]. Thus, more benign and sustainable strategy, such as utilization of renewable resources as raw materials to economically produce benzene, is highly desired, which can liberate us from the reliance on fossil resource and is of great industrial and social significance[12,13]. Such an attractive, cheap and non-edible material is lignocellulosic biomass, generated from forestry and agricultural activity worldwide[14]. As a main constituent of lignocellulose, lignin is the most abundant renewable source composed of aromatic building blocks in nature[15], with an annual production of around 50 billion metric tons[16]. In terms of molecular structure, the aromatic character of lignin springs from the benzene ring structures, which renders itself a sustainable and desirable candidate feedstock for benzene production[16–18].

In recent years, valorizing lignin into fossil-based chemicals has received extensive attention, and various valuable chemicals used for transport fuels and industrial manufactures have been

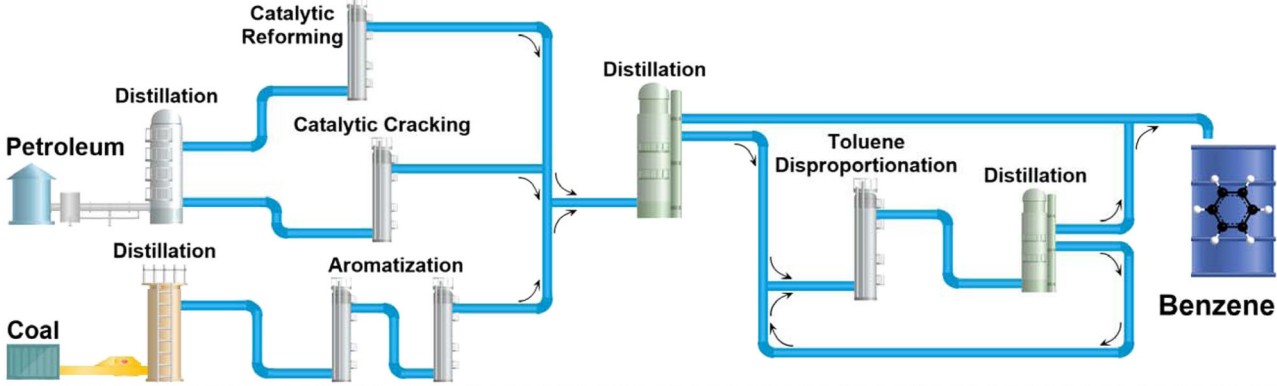

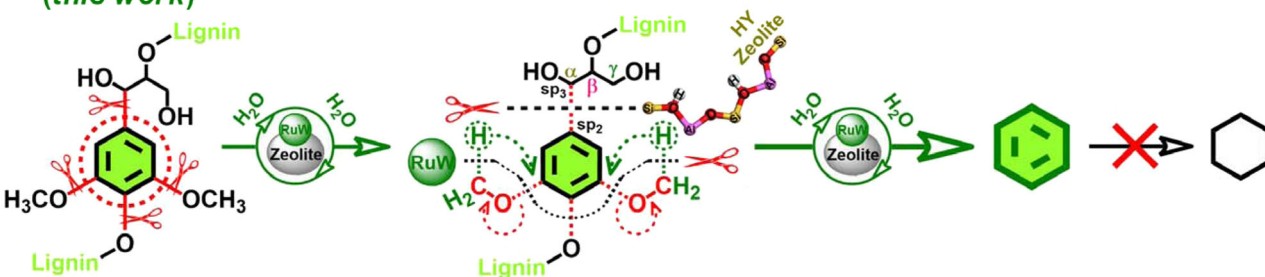

**Fig. 1 Strategies employed for the benzene production. a** Traditional industry route for benzene production. **b** Natural gas route for benzene production. **c** Lignin-to-benzene route.

obtained employing lignin as the feedstock[19–28]. For example, advances in the seminal work of Luterbacher and co-workers on aldehyde stabilization of lignin improved the lignin depolymerization efficiency and yielded the mixture of guaiacyl and syringyl monomers, such as alkyl-substituted guaiacol/syringol[20,21], hydroxypropyl-substituted guaiacol/syringol[20,21] and guaiacyl/syringyl propane diones[22], via hydrogenolysis and oxidation approaches. In the pioneering work of Stahl and co-workers[23], lignin could be depolymerized into syringaldehyde and vanillin via successive oxidation-formylation-hydrolysis processes. Wang and co-workers conducted the direct hydrodeoxygenation of lignin to alkyl-substituted arenes and hydrocarbons[24]. Sels and co-workers[25] developed an integrated biorefinery process that converted lignin into phenol with high yield. Wang and co-workers[26] also obtained phenol product from lignin by a multistep oxidation and decarboxylation methodology. These findings place a premium on strategies that pursue benzene from lignin, however in the currently reported methodologies, such as catalytic pyrolysis[29–31], hydrodeoxygenation[32–34] and combined catalytic processing[15,35], benzene could only be detected in quite a low yield in the complex mixture containing the aforementioned phenolic hydroxyl, methoxyl and alkyl-substituted aromatic products. Although there is ongoing research on lignin valorization, efficient lignin-to-benzene route has not been reported so far. Obviously, the complex chemical bonding environment, especially the stable aromatic carbon–aliphatic carbon ($C_{sp2}$–$C_{sp3}$) and aromatic carbon–oxygen ($C_{sp2}$–O) bonds functionalized on the benzene rings truly limited the abstraction of benzene from the lignin structure. In this context, further cleavage of the unbroken $C_{sp2}$–$C_{sp3}$/$C_{sp2}$–O bonds in the above generated alkylbenzenes and alkylphenols for desired benzene product can only be realized via dealkylation, dealkenylation and reductive cleavage processes under quite harsh conditions (for example, high temperature, etc.)[36–40]. Moreover, such required reaction conditions often lead to inevitable side reactions[33]. For example, the concurrent hydrogenation of the benzene ring in the hydroprocessing stage of the lignin upgradation with exogenous hydrogen source always further reduces the benzene selectivity[24,41,42]. Thus, a key and great challenge for efficient lignin-to-benzene route would be to design practical strategy for combining the complicated refinements of the $C_{sp2}$–$C_{sp3}$/$C_{sp2}$–O bonds into $C_{sp2}$–H bonds via friendly catalysis, without multistep technological procedure and extra hydrogenation of benzene ring. In our previous work, we reported the hydrogenolysis of the aromatic $C_{sp2}$–O($CH_3$) bond without need of any exogenous hydrogen or other reductants, termed as self-supported hydrogenolysis (SSH, Supplementary Fig. 2), by which the $C_{sp2}$–O bond can be transformed into $C_{sp2}$–H bond using the in situ abstracted hydrogen from the methoxyl group, and the side hydrogenation of the benzene ring can be completely avoided[43]. Besides, some catalysts, such as $MoO_3$, have also been proven to be effective in hydrodeoxygenation process of the lignin model compounds or lignin oil for arene products with high selectivity, using exogenous hydrogen[32,44].

To develop desired lignin-to-benzene route, in this work, we propose an integrated catalytic strategy that in situ refines the $C_{sp2}$–$C_{sp3}$/$C_{sp2}$–O bonds functionalized on the benzene ring structures into $C_{sp2}$–H bonds over the RuW/$HY_{30}$ catalyst without any saturation of the ring, thereby achieving a single production of benzene from lignin (Fig. 1c). The in situ refining mechanism of the $C_{sp2}$–$C_{sp3}$/$C_{sp2}$–O bonds is disclosed, which effectively couples the direct deconstruction of the $C_{sp2}$–$C_{sp3}$ bonds in the phenylpropanol units on the local structure of the lignin molecule without any precedent reductive fractionation process, and the hydrogenolysis of the $C_{sp2}$–O bonds using the local-access hydrogen source in lignin molecule. With water

acting as the cheap and green reaction medium, no other reductant or reactant is needed. This efficient strategy affords mild-condition abstraction of benzene from lignin molecule structures with a benzene yield as high as 18.8% on lignin weight basis, and our high-performance protocol can be readily scaled up to produce 8.5 g of benzene product from 50.0 g lignin without any saturation byproducts.

## Results

**Optimization study.** We focused our in situ refining strategy studies initially on the overall transformation of the $C_{sp2}$–$C_{sp3}$/$C_{sp2}$–O bonds by using 1-(4-methoxyphenyl)-1-propanol (**1a**) as the model compound because of its lignin-mimetic phenylpropanol structure [($CH_3O$)Ph-$C_{sp3}$(OH)–] with only one methoxy group [$C_{sp2}$–O($CH_3$)] and one 1-hydroxypropyl group [$C_{sp2}$–$C_{sp3}$(OH)] substituted on the benzene ring, which simplifies the study. After extensive screening of various conditions, we identified the optimized catalytic system and reaction conditions shown in Table 1 and Supplementary Fig. 3, which employed cheap and commercially available hydrogen-type faujasite Y zeolite ($HY_{30}$, Si/Al ratio = 30) with supported RuW alloy component using water as the reaction medium at 180 °C under nitrogen atmosphere (Table 1, entries 1–3, Supplementary Fig. 3a, b). As expected, the transformations of the $C_{sp2}$–$C_{sp3}$/$C_{sp2}$–O bonds in **1a** did not occur without the catalyst, despite the achievable dehydroxylation (**1h**) in the blank experiment (Table 1, entry 1)[45]. It is clear that $HY_{30}$ zeolite efficiently catalyzed the deconstruction of the $C_{sp2}$–$C_{sp3}$ bond in **1a** into $C_{sp2}$–H bond, but unfortunately the $C_{sp2}$–O bond in anisole (**1d**) was not reactive when only $HY_{30}$ was used as the catalyst (Table 1, entry 2). With RuW alloy introduced into the catalytic system, these unreacted anisole molecules could be further converted into benzene (**1b**) via the SSH process of the $C_{sp2}$–O bond (Table 1, entry 3). It can be found that the selectivity to benzene (**1b**) could be 97.3% with a remained anisole (**1d**) selectivity of 2.5% at 99.9% conversion of **1a** over the RuW/$HY_{30}$ catalyst under nitrogen atmosphere in water (Table 1, entry 3, Supplementary Fig. 3b, c), and according to the $^1H$ nuclear magnetic resonance (NMR) analysis of the aqueous phase (Supplementary Fig. 3d), there was no water-soluble product. As the reaction medium in the catalytic system, water also served as an ideal booster for the SSH reaction of the $C_{sp2}$–O bond by way of forming hydrogen bonds with the O atom in anisole[46,47], which was beneficial for the in situ refining strategy. A total conversion of **1a** into benzene product could be achieved in 6 h without any saturated cyclohexane product (**1c**) (Table 1, entry 4). In contrast, a test with exogenous hydrogen source led to benzene and cyclohexane products with competitive selectivities of 43.3% and 16.9%, which was additionally coupled with n-propylbenzene (**1e**) and n-propylcyclohexane (**1f**) in respective selectivities of 35.8% and 3.9% (Table 1, entry 5), albeit in lowered selectivities of cyclohexane (**1c**), n-propylbenzene (**1e**) and n-propylcyclohexane (**1f**) under low pressure of exogenous $H_2$ (Table 1, entry 6). This is rationalized by the inevitable competitions of the hydrogenation of the benzene rings and hydrogenolysis of the hydroxyl group substituted at the aliphatic α-C ($C_\alpha$) position (namely the $C_{sp3}$ carbon in the $C_{sp2}$–$C_{sp3}$ bond, Fig. 1c) in the traditional hydroprocessing processes using exogenous hydrogen as reductant[14], which have been successfully avoided via the $HY_{30}$ zeolite and RuW NPs accomplished refining strategy for the in situ transformation of the $C_{sp2}$–$C_{sp3}$ and $C_{sp2}$–O bonds. To confirm the key roles of RuW and $HY_{30}$ in the catalysts, we prepared the reference Ru/$HY_{30}$, W/$HY_{30}$, RuW/$SiO_2$ and RuW/$Al_2O_3$ catalysts, and studied their catalytic performances for the reaction of **1a** (Table 1, entries 7–10). Under nitrogen atmosphere, the

**Table 1 Screening of catalysts for in situ refining 1-(4-methoxyphenyl)-1-propanol (1a) over supported metal catalysts at different conditions.**

| Entry | Catalytic system[a] Catalyst[b] | Gas | t (h) | Conv. (%) | Select. (%) 1b | 1c | 1d | 1e | 1f | 1g | 1h | Yie. of 1b (%) |
|---|---|---|---|---|---|---|---|---|---|---|---|---|
| 1[c] | – | $N_2$ | 5.5 | 99.9 | 0.0 | 0.0 | 0.0 | 0.0 | 0.0 | 0.0 | 99.9 | 0.0 |
| 2 | $HY_{30}$ | $N_2$ | 1.0 | 99.9 | 0.0 | 0.0 | 99.0 | 0.0 | 0.0 | 0.0 | 0.0 | 0.0 |
| 3 | $RuW/HY_{30}$ | $N_2$ | 5.5 | 99.9 | 97.3 | 0.0 | 2.5 | 0.0 | 0.0 | 0.0 | 0.0 | 97.2 |
| 4 | $RuW/HY_{30}$ | $N_2$ | 6.0 | 99.9 | 99.6 | 0.0 | 0.0 | 0.0 | 0.0 | 0.0 | 0.0 | 99.5 |
| 5[d] | $RuW/HY_{30}$ | $H_2$ | 6.0 | 99.9 | 43.3 | 16.9 | 0.0 | 35.8 | 3.9 | 0.0 | 0.0 | 43.2 |
| 6[e] | $RuW/HY_{30}$ | $H_2$ | 6.0 | 99.9 | 86.6 | 2.0 | 0.0 | 10.8 | 0.5 | 0.0 | 0.0 | 86.5 |
| 7 | $Ru/HY_{30}$ | $N_2$ | 5.5 | 99.9 | 0.0 | 0.0 | 98.8 | 0.0 | 0.0 | 0.0 | 0.0 | 0.0 |
| 8 | $W/HY_{30}$ | $N_2$ | 5.5 | 99.9 | 0.0 | 0.0 | 98.6 | 0.0 | 0.0 | 0.0 | 0.0 | 0.0 |
| 9[f] | $RuW/SiO_2$ | $N_2$ | 5.5 | 99.9 | 0.0 | 0.0 | 0.0 | 94.5 | 0.0 | 4.9 | 0.0 | 0.0 |
| 10[g] | $RuW/Al_2O_3$ | $N_2$ | 5.5 | 99.9 | 0.0 | 0.0 | 0.0 | 72.2 | 0.0 | 9.5 | 0.0 | 0.0 |
| 11 | $RuW/HY_3$ | $N_2$ | 5.5 | 99.9 | 10.8 | 0.0 | 0.0 | 73.0 | 0.0 | 5.0 | 0.0 | 10.8 |
| 12 | $RuW/HY_5$ | $N_2$ | 5.5 | 99.9 | 13.2 | 0.0 | 0.0 | 73.3 | 0.0 | 4.3 | 0.0 | 13.2 |
| 13 | $RuW/HY_{15}$ | $N_2$ | 5.5 | 99.9 | 31.9 | 0.0 | 0.0 | 61.8 | 0.0 | 2.1 | 0.0 | 31.9 |
| 14[h] | $RuW/HY_{40}$ | $N_2$ | 5.5 | 99.9 | 96.0 | 0.0 | 3.5 | 0.0 | 0.0 | 0.0 | 0.0 | 95.9 |
| 15[i] | RuW/Beta | $N_2$ | 5.5 | 99.9 | 10.0 | 0.0 | 0.0 | 75.0 | 0.0 | 3.5 | 0.0 | 10.0 |
| 16 | $RuW/HZSM-5_{14}$ | $N_2$ | 5.5 | 99.9 | 8.9 | 0.0 | 0.0 | 70.0 | 0.0 | 8.3 | 0.0 | 8.9 |
| 17 | $RuW/HZSM-5_{18}$ | $N_2$ | 5.5 | 99.9 | 8.1 | 0.0 | 0.0 | 74.9 | 0.0 | 7.7 | 0.0 | 8.1 |
| 18 | $RuW/HZSM-5_{30}$ | $N_2$ | 5.5 | 99.9 | 7.5 | 0.0 | 0.0 | 78.1 | 0.0 | 7.1 | 0.0 | 7.5 |
| 19 | $RuW/HZSM-5_{35}$ | $N_2$ | 5.5 | 99.9 | 6.3 | 0.0 | 0.0 | 82.6 | 0.0 | 6.2 | 0.0 | 6.3 |
| 20 | $RuW/HZSM-5_{65}$ | $N_2$ | 5.5 | 99.9 | 5.2 | 0.0 | 0.0 | 87.6 | 0.0 | 4.9 | 0.0 | 5.2 |
| 21 | $RuW/HZSM-5_{235}$ | $N_2$ | 5.5 | 99.9 | 2.0 | 0.0 | 0.0 | 92.9 | 0.0 | 3.0 | 0.0 | 2.0 |
| 22[j] | RuW/Mordenite | $N_2$ | 5.5 | 99.9 | 0.0 | 0.0 | 0.0 | 71.6 | 0.0 | 8.2 | 0.0 | 0.0 |
| 23[k] | RuW/SAPO | $N_2$ | 5.5 | 99.9 | 0.0 | 0.0 | 0.0 | 83.8 | 0.0 | 9.2 | 0.0 | 0.0 |
| 24[l] | RuW/MCM-41 | $N_2$ | 5.5 | 99.9 | 0.0 | 0.0 | 0.0 | 94.0 | 0.0 | 5.1 | 0.0 | 0.0 |

[a]Reaction results are the averages of three experiments conducted in parallel. 1a (1.0 mmol), $H_2O$ (5.0 mL), 180 °C, 0.1 MPa $N_2$, 800 rpm.
[b]Ru/$HY_{30}$ (0.18 g, 2.5 wt% Ru) and W/$HY_{30}$ (0.20 g, 14 wt% W) catalysts (0.20 g, 2.5 wt% Ru, 14 wt% W, the content of metal is based on zeolite, $SiO_2$ and $Al_2O_3$ materials, and determined by ICP); RuW/zeolite, RuW/$SiO_2$ and RuW/$Al_2O_3$ catalysts (0.20 g, 2.5 wt% Ru, 14 wt% W, the content of metal is based on $HY_{30}$ zeolite and determined by ICP).
[c]Without catalyst.
[d]1.0 MPa $H_2$.
[e]0.1 MPa $H_2$.
[f]Amorphous $SiO_2$.
[g]γ-$Al_2O_3$.
[h]The catalyst dosage is 1.5 times that of the RuW/$HY_{30}$ catalyst.
[i]Hydrogen-type Beta, Si/Al ratio is 25.
[j]Hydrogen-type Mordenite, Si/Al ratio is 10.
[k]Hydrogen-type SAPO-34, Si/Al ratios is 0.25.
[l]All-silicon MCM-41.

monometallic Ru/HY$_{30}$ (Table 1, entry 7) and W/HY$_{30}$ (Table 1, entry 8) catalysts performed the same function as HY$_{30}$ zeolite, and just led to the anisole product. Likewise, the bimetallic MW/HY$_{30}$ (M = Ni, Co, Fe, Mo, and Cu) catalysts (Supplementary Table 1, entries 1–5) also only catalyzed the C$_{sp2}$–C$_{sp3}$ bond deconstruction without reaction of C$_{sp2}$–O bond, proving that the combination of Ru and W was necessary for the SSH reaction of the C$_{sp2}$–O bond. Although the C$_{sp2}$–O bond could be hydrogenolyzed in the presence of the exogenous hydrogen over the MW/HY$_{30}$ catalysts (for example, NiW/HY$_{30}$, CoW/HY$_{30}$, and FeW/HY$_{30}$), their selectivities of benzene were much lower than that over RuW/HY$_{30}$ catalyst (Supplementary Table 1, entries 6–10). As the constituent of HY$_{30}$ zeolite, silicon and aluminum oxides supported RuW catalysts (Table 1, entries 9 and 10) only promoted the conversion of 1a to n-propylbenzene (1e) without any benzene product, disclosing that the unique catalytic properties derived from the zeolitic aluminosilicate [–TO$_4$–, (T = Si, Al)] frameworks can offer a particular way to tailor the C$_{sp2}$–C$_{sp3}$ bond into C$_{sp2}$–H bond under mild conditions. The above control experiments showed that the HY$_{30}$ zeolite and RuW alloy are respectively essential to realizing the transformations of the C$_{sp2}$–C$_{sp3}$ and C$_{sp2}$–O bonds into C$_{sp2}$–H bonds for producing benzene product (1b).

To get insight into the structural origins of the observed excellent catalytic performance, we characterized the catalysts by several techniques. A typical x-ray diffraction (XRD) pattern of the as-synthesized RuW/HY$_{30}$ catalyst (Fig. 2a) revealed well-defined diffraction peaks of the RuW alloy and HY$_{30}$ zeolite structures. Based on the Rietveld refinement analysis of the XRD data (Fig. 2b), the random occupancies of Ru and W atoms are estimated to be 20.3 and 79.7% in the RuW nanoparticles (NPs) with a range of 1.0–4.0 nm (Supplementary Fig. 4a–c). The Ru K-edge extended x-ray absorption fine structure (EXAFS) fitting (Fig. 2c and Supplementary Table 2) exhibits a Ru-W first shell coordination number (CN) of 3.1, while the W-Ru CN was just 0.9 due to the high W/Ru molar ratio (Fig. 2d and Supplementary Table 2), directly proving the alloy properties between the homogeneously distributed Ru and W atoms in the HY$_{30}$ supported RuW NPs (Supplementary Fig. 4d–i). Accordingly, X-ray photoelectron spectroscopy (XPS) analysis of the RuW/HY$_{30}$ catalyst showed the opposite shifts for the Ru3p (towards lower energy, Fig. 2e) and W4f (towards higher energy, Fig. 2f) orbital peaks, as compared respectively to the Ru/HY$_{30}$ and W/HY$_{30}$ catalysts, which can be ascribed to that the higher electronegativity of Ru induced electron transfer from W to Ru in the RuW NPs[47].This interplay between Ru and W atoms could also be discovered from the slightly lower energy absorption threshold value in the Ru K-edge x-ray absorption near-edge structure (XANES) curve of the RuW/HY$_{30}$ catalyst (Supplementary Fig. 5a), as compared with that of Ru foil, despite the shift tendency to higher threshold value was inconspicuous for W L$_3$-edge (Supplementary Fig. 5b). The analyses of the acid and textural properties indicated that the RuW/HY$_{30}$ catalyst was highly strong Bronsted acidic (Supplementary Table 3), and had substantial mesoporous structures which are favorable to the access of reactant molecules to the acid sites, and then the rapid deconstruction of the C$_{sp2}$–C$_{sp3}$ bonds (Supplementary Table 4, Supplementary Fig. 4b, c). Compared with RuW/HY$_{30}$ catalyst, although the RuW/HY catalysts with lower zeolite framework Si/Al ratios (3, 5, and 15) possessed more acid sites, their strong Bronsted acid sites were in lower proportions (Supplementary Table 3) and actually distributed in the micropores with higher proportions (Supplementary Table 4), which heavily restricted the deconstruction efficiency of the C$_{sp2}$–C$_{sp3}$ bonds on the Bronsted acid sites due to the inferior diffusibility of the microporous structures, matching that only less than one-third of 1a were

transformed into benzene, and the main product was n-propylbenzene (1e) (Table 1, entries 11–13). RuW/HY$_{40}$ catalyst with a higher zeolite Si/Al ratio of 40 had the appropriate textural properties, which gave a high selectivity of benzene (Table 1, entry 14), but requiring 1.5 times catalyst dosage of RuW/HY$_{30}$ catalyst, due to its lower quantity of Bronsted acid sites (Supplementary Table 3). The overall transformation of 1a could also be achieved over other types of zeolite supported RuW catalysts with inferior mesoporous structures (Table 1, entries 15–23), but similarly generate n-propylbenzene (1e) in quantity, albeit with limited selectivities of benzene over RuW/Beta (Table 1, entry 15) and RuW/HZSM-5 (Table 1, entries 16–21) catalysts. It is well known that pure silica zeolites do not contain any acid sites[48], thus RuW/MCM-41 catalyst was incapable of C$_{sp2}$–C$_{sp3}$ bond deconstruction, also with n-propylbenzene (1e) as the main product (Table 1, entry 24). As the comparison of the utility of the RuW/HY$_{30}$ catalyst, it is the poor activities of the supports in the above catalysts (Table 1, entries 9–13 and 15–24) on the deconstruction of the C$_{sp2}$–C$_{sp3}$ bonds that offered RuW the opportunity to hydrogenolyze the hydroxyl group substituted at the aliphatic C$_{sp3}$ position of the [C$_{sp2}$–C$_{sp3}$(OH)] motif using the active hydrogen derived from the formaldehyde molecules generated during the SSH reaction of the C$_{sp2}$–O bonds (Supplementary Fig. 2), consequently leading to the n-propylbenzene byproduct (Supplementary Table 5). Similarly, the 1-methoxy-4-propylbenzene (1g) was also resulted from the above hydrogenolysis of the hydroxyl group in [C$_{sp2}$–C$_{sp3}$(OH)] motif of 1a, but without SSH reaction of the C$_{sp2}$–O bond proceeding yet. Aside from the catalytic properties, RuW/HY$_{30}$ catalyst also exhibited excellent stability in the reaction, which was confirmed by reusing the catalyst (Supplementary Fig. 6a) and characterizations of the catalyst before and after the reaction by XRD (Supplementary Fig. 6b) and TEM techniques (Supplementary Fig. 6c).

**Mechanistic study**. On the basis of the above comparison tests (Table 1), and given that traditional hydroprocessing with exogenous hydrogen source resulted not only in the unavoidable hydrogenation of the benzene ring to cyclohexane but also in the n-propylbenzene byproduct that could not be further converted into benzene under the mild condition (Table 1, entries 5 and 6), we proposed that the outstanding efficiency of the refining strategy for benzene should be attributed to the innovative matching of RuW NPs and HY$_{30}$ zeolite in the catalyst, where RuW NPs catalyzed hydrogenolysis of the C$_{sp2}$–O bonds with the in situ abstracted hydrogen from the reactant can not only completely avoid the saturation of the benzene ring, but more importantly, allow the C$_{sp2}$–C$_{sp3}$ bonds to deconstruct promptly over the HY$_{30}$ zeolite without the competition from the hydrogenolysis of the hydroxyl group in [C$_{sp2}$–C$_{sp3}$(OH)] motif. The pathway for the SSH reaction of C$_{sp2}$–O bond on the RuW centers has been established and illustrated in Supplementary Fig. 2[43]. We here focus on the study of the unique catalysis of C$_{sp2}$–C$_{sp3}$ into C$_{sp2}$–H bond by means of the HY$_{30}$ zeolite. Based on our experimental characterizations, we constructed the Bronsted acid model to represent the C$_{sp2}$–C$_{sp3}$ reaction center in the RuW/HY$_{30}$ catalyst, and performed density functional theory (DFT) calculations to study the reaction path of the C$_{sp2}$–C$_{sp3}$ bond. As shown in Fig. 3a, initially, the hydroxyl group substituted at the aliphatic C$_\alpha$ position is easily protonated on the Bronsted acid site, leading to the oxonium ion (structure I) that then transforms into a carbonium ion (structure II) by eliminating a molecule of H$_2$O. Thermodynamically, the γ-methyl on the side chain is then shifted to the C$_\alpha$ position under the catalysis of the Bronsted acid center, evolving to the chemically adsorbed

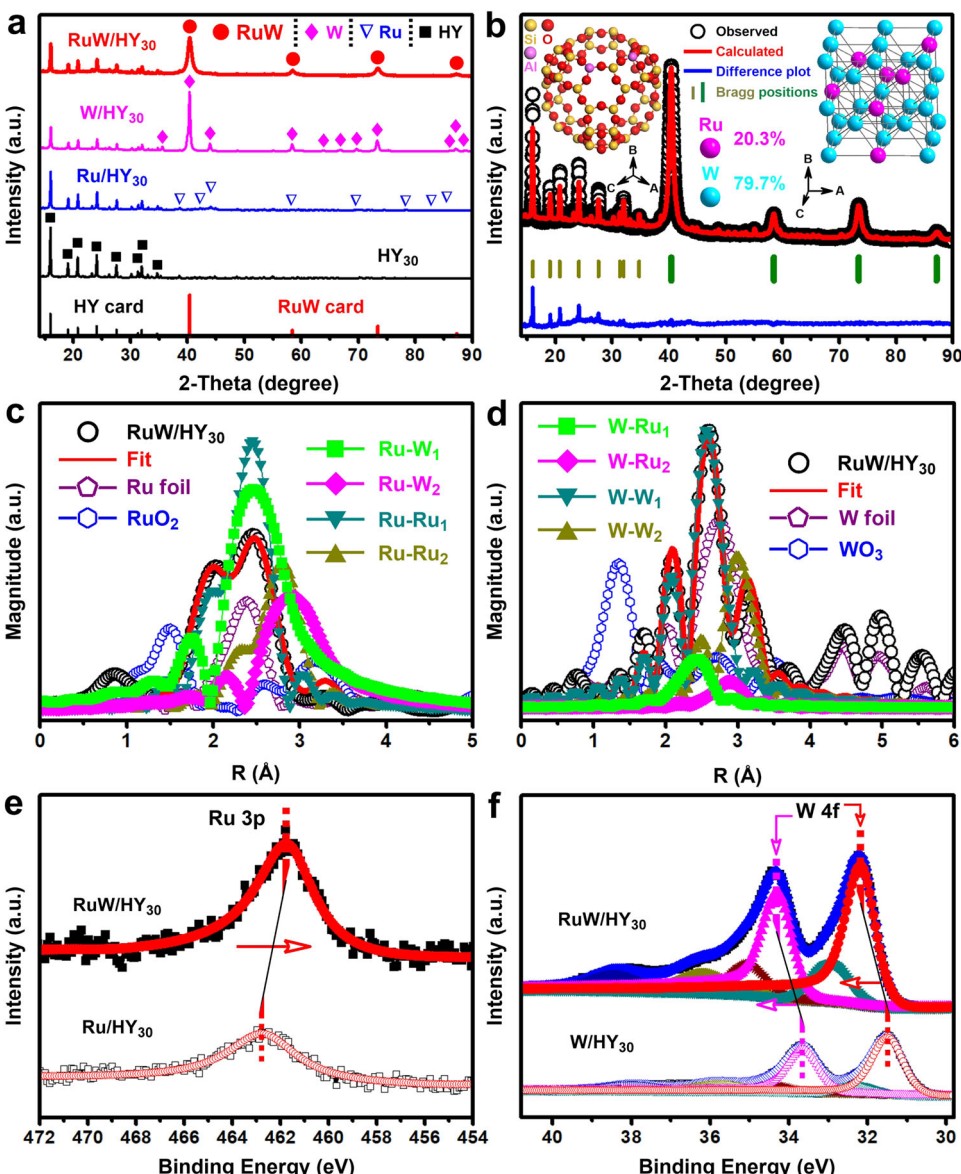

**Fig. 2 Characterization of RuW/HY$_{30}$ catalyst. a** XRD patterns of RuW/HY$_{30}$ (2.5 wt% Ru, 14 wt% W) catalyst (red), W/HY$_{30}$ (14 wt% W) catalyst (pink), Ru/HY$_{30}$ (2.5 wt% Ru) catalyst (blue) and HY$_{30}$ (black), standard patterns of RuW alloy and Y zeolite from ICDD PDF cards (65–6705 for RuW, 45–0112 for Y zeolite) are shown at the bottom. **b** Rietveld refinement of RuW/HY$_{30}$ catalyst (2.5 wt% Ru, 14 wt% W) using XRD pattern in **a** (inset: crystal structures of RuW alloy NPs and Y zeolite). Black circle marks (○) represent the observed intensities, and the red solid line is Rietveld-fit profile. The difference plot (blue) is shown at the bottom. The dark yellow (Y zeolite) and olive (RuW alloy) tick marks respectively indicate the positions of the Bragg reflections as obtained in the Rietveld refinement. The RuW alloy parameters are as follows: space group $Im$-$3m$, a = b = c = 3.1610 Å. $R_p$ = 7.37% and $R_{wp}$ = 9.31%. EXAFS Fourier transformed (FT) $k^2$-weighted $\chi(k)$ function spectra of the RuW/HY$_{30}$ catalyst, references, and corresponding EXAFS $R$-space fitting curves for Ru (**c**) and W (**d**) species respectively in the RuW/HY$_{30}$ catalyst. XPS spectra for Ru 3p (**e**) and W 4f regions (**f**).

structure III with a migration barrier of 1.25 eV (TS-1), and subsequent proton abstraction by the Bronsted acid center delivers structure IV in a lower barrier of 0.89 eV (TS-2). Then, by a second protonation at the C$_\alpha$ position of structure V, the resulting carbonium ion V (structure V) would be formed by the trap of H$_2$O, which gives tertiary alcohol geometry (structure VI) and concomitantly regenerate the Bronsted acid center with the proton. When the benzene ring of the tertiary alcohol (structure VI) is turned endothermically toward the Bronsted acid center, an appropriate geometry (structure VII) is formed to enable the protonation of the C$_{sp2}$ position (structure VIII) with an effective barrier of 0.53 eV (TS-3). After that, the C$_{sp2}$−C$_{sp3}$ bond can be exothermically transformed into the desired C$_{sp2}$−H bond via the ultimate β scission step[36].

Taken together, the mild-condition refining of C$_{sp2}$−C$_{sp3}$ bond into C$_{sp2}$−H bond was proceeded via a stepwise protonated dehydroxylation, γ-methyl shift and C$_{sp2}$−C$_{sp3}$ β scission pathway. Although the dehydroxylation could also occur in the water-only system (Table 1, entry 1), 1-methoxy-4-(prop-1-en-1-yl) benzene (**1h**) was the sole product without other following steps, which provided compelling proof for the requisite catalysis worked in the desired refining of the C$_{sp2}$−C$_{sp3}$ bond. In terms of catalytic chemistry, highly active acid nature and chemical accessibility of the framework in the mesopore-rich HY$_{30}$ innately enabled the aliphatic γ-methyl C (C$_\gamma$) atom to be activated on the Bronsted acid center, which could be illustrated by the spacing-filling models (Fig. 3b), affording the migration of the γ-methyl[49,50]. Such a chemical bonding state should be

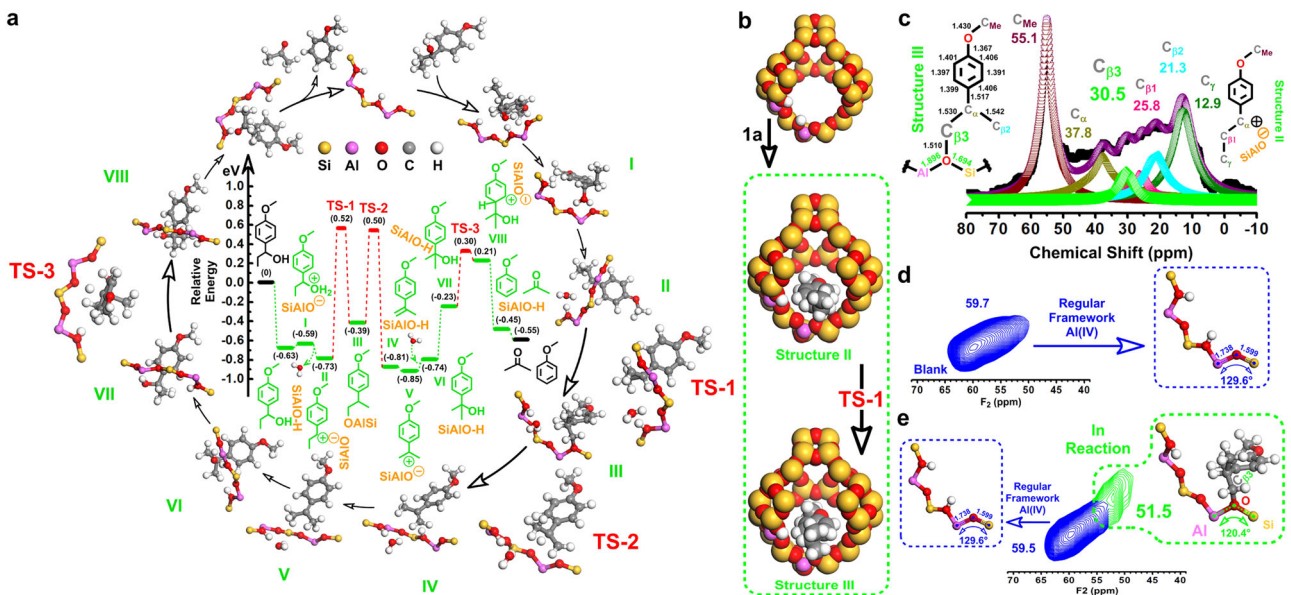

**Fig. 3 Mechanistic study for the transformation of the $C_{sp2}$–$C_{sp3}$ bond. a** Energy profile for transformation of the $C_{sp2}$–$C_{sp3}$ bond. Reaction intermediates (**I**–**VIII**) and transition states (**TS**, **1**–**3**). [SiAlO], the Bronsted acid center in the RuW/HY$_{30}$ catalyst. **b** Space-filling models of the Bronsted acid center and intermediate structures (Structures II and III) in **a**. **c** Core bond metrics of structure III and $^{13}$C NMR analysis of the C–C bond evolution in reaction of **1a**, black solid square (■) represents the observed intensities, purple circle marks (●) is the total fitting curve, dark yellow regular triangle upward (△), pink right triangle (▷), cyan hollow left triangle (◁), green diamond (◇), olive hollow hexagon (⬡) and wine regular triangle downward (▽) represent the fitting curves of $C_\alpha$, $C_{\beta1}$, $C_{\beta2}$, $C_{\beta3}$, $C_\gamma$ and $C_{Me}$, respectively. Reaction conditions: **1a** (1 mmol), RuW/HY$_{30}$ catalyst (0.20 g, 2.5 wt% Ru, 14 wt% W), H$_2$O (5.0 ml), 180 °C, 10 min, 0.1 MPa N$_2$, 800 rpm. **d** $^{27}$Al MQMAS analysis of the boiled RuW/HY$_{30}$ catalyst in the blank experiment, blue loops represent the regular framework Al(IV), F2, Fourier transformation in the directly detected dimension. Reaction conditions: RuW/HY$_{30}$ catalyst (0.20 g, 2.5 wt% Ru, 14 wt% W), H$_2$O (5.0 ml), 180 °C, 10 min, 0.1 MPa N$_2$, 800 rpm. **e** $^{27}$Al MQMAS analysis of the RuW/HY$_{30}$ catalyst in reaction of **1a**, blue and green loops represent the regular framework Al(IV) and the framework Al(IV) in the [Al–O($C_{\beta3}$)–Si] center (Structure III, **a**), respectively. Reaction conditions are the same as that in **c**.

accompanied by a change in the chemical environment of the Bronsted acid center and reactant, which were monitored by solid-state magic angle spinning nuclear magnetic resonance analyses of the quenched catalytic system, as well as the corresponding model compounds (Fig. 3c–e, Supplementary Figs. 7–10, Supplementary Tables 6 and 7). The $^{13}$C NMR analysis showed a clear resonance signal at 55.1 ppm, characteristic of methyl C atom in the methoxy group, and the dehydroxylated $C_\alpha$ atom shift of 37.8 ppm instead of the initial $C_\alpha$ shift at 73.8 ppm (Fig. 3c, Supplementary Fig. 7 and Supplementary Table 6), which are indicative of the catalysis-priority of the Bronsted acid center during the reaction of **1a**, evidencing the trend of the product distribution on reaction time (Supplementary Fig. 3b). Although the $^{13}$C NMR signal for the $C_\gamma$ atom could still be detected, a new resonance signal at 30.5 ppm was also observed, together with the 25.8 and 21.3 ppm signals for the original β1-C ($C_{\beta1}$, in Structure II) and β2-C ($C_{\beta2}$, in Structure III) atoms, which could be ascribed to the β3-C ($C_{\beta3}$, in Structure III) atom shifted from the $C_\gamma$ atom in reaction[51]. The above resolved signals were further supported by the solid-state two-dimensional (2D) $^{13}$C{$^1$H} dipolar-mediated heteronuclear correlation (HETCOR) NMR analysis (Supplementary Fig. 8 and Supplementary Table 7). The original $C_{\beta1}$ and $C_{\beta2}$ atoms at 25.3 and 21.8 ppm are strongly correlated with the $^1$H signals at 1.4 and 1.8 ppm that should be associated with (–$C_{\beta1}H_2$–) and (–$C_{\beta2}H_3$) moieties in the intermediates II and III, respectively. The observed $^{13}$C and $^1$H signals at 13.5 and 1.0 ppm are assigned to the remained (–$C_\gamma H_3$) moiety in the intermediate II. In line with the $^{13}$C NMR analysis (Fig. 3c), 2D $^{13}$C{$^1$H} HETCOR NMR also yields a well-resolved correlated signal at 31.0 ppm in the $^{13}$C dimension and at 3.0 ppm in the $^1$H dimension, which again

reflects the transferred $C_\gamma$, namely, new $C_{\beta3}$ atom from the (–$C_{\beta3}H_2$–SiAlO) moiety in the intermediate III. It is the chemical bonding with the oxygen atom in the Bronsted acid [≡Al–O–Si≡] center that provided an inductive effect on the $C_{\beta3}$ atom, which shifted its resonance downfield from that of the original $C_{\beta1}$ and $C_{\beta2}$ atoms, precisely supporting the Bronsted acid-catalyzed mechanism of the γ-methyl shift[52,53]. Echoing the chemical environment change of the reactant, such a chemical bonding in the [≡Al–O($C_{\beta3}$)–Si≡] unit also fed back to the tetrahedrally coordinated Al [Al(VI)] in the Bronsted acid center, which was further confirmed by $^{27}$Al multiple-quantum MAS (MQMAS) (Fig. 3d, e, Supplementary Fig. 9) and $^{27}$Al NMR (Supplementary Fig. 10) analyses. In the blank experiment (without reactant, Fig. 3d), the regular framework Al(VI) signal in the 2D plot of the boiled catalyst is clearly evident at 59.7 ppm (F2, Fig. 3d and Supplementary Fig. 9a)[54]. Besides the above regular Al(VI) signal, as expected, a resonance signal at 51.5 ppm (F2) was distinctly identified in the operating RuW/HY$_{30}$ catalyst (Fig. 3e and Supplementary Fig. 9b), which had also been detected as a shoulder signal in the $^{27}$Al NMR results (Supplementary Fig. 10). This emerging Al environment could be assigned to the Al(VI) atom in the distorted [≡Al–O($C_{\beta3}$)–Si≡] unit, where the charge compensation of the $C_{\beta3}$ atom weakened the inductive effect of the O atom on the neighboring Al(VI) atom, and then reduced the Al–O–Si bond angel and lengthened the Al–O and Si–O bonds (Fig. 3e and Supplementary Fig. 10b), shifting the resonance upfield from that of the framework Al(VI) atoms in the regular [≡Al–O–Si≡] environments[55]. The above identifications evidently described the key processes that occurred on the $C_{sp2}$–$C_{sp3}$ refining centers of the RuW/HY$_{30}$ catalyst, highly agreeing with the DFT mechanism studies (Fig. 3a).

**Scope of phenylpropanol derivatives**. With the optimized conditions established, we proceeded to study the scope of the in situ refining strategy. As shown in Fig. 4, several structurally and substitutionally distinct $C_{sp2}-C_{sp3}/C_{sp2}-O$ bonds equipped phenylpropanol derivatives were refined effectively and singly into desired benzene product, including those G (guaiacyl)-monomeric lignin model compounds (Fig. 4a) bearing methoxyl (**2a**), hydroxyl (**3a**) and isopropoxyl (**4a**) groups, as well as the S (syringyl)-monomers (Fig. 4b, **5a**, **6a** and **7a**). The in situ refining strategy is not limited to the monomers, a range of G and S-monomer derived dimeric model compounds including those bearing the methoxyl groups (Fig. 4c, **8a** and **9a**) and the combinatorial groups of methoxyl and isopropoxyl (Fig. 4d, **10a–13a**), were all viable in the reactions with benzene as the single product. The catalytic performance observed here represented an important proof of the multifunctional centers on the RuW/HY$_{30}$ catalyst, where the interlaced $C_{sp2}-C_{sp3}$ and $C_{sp2}-O$ bonds in the complex compounds could be efficiently tolerated and transformed into the $C_{sp2}-H$ bonds, respectively, demonstrating the feasibility of the in situ refining strategy (Fig. 1c). As a further demonstration of the in situ refining protocol, we further performed the detailed analysis of the conversion of 1-(4-isopropoxy-3, 5-dimethoxyphenyl)-2-(2-methoxyphenoxy)propane-1, 3-diol (**11a**) versus reaction time, which aimed to comprehensively understand the evolution of the $C_{sp2}-C_{sp3}/C_{sp2}-O$ bonds in the lignin-mimetic dimeric linkages (Supplementary Fig. 11). When **11a** was subjected to our reaction conditions, the transformation proceeded readily to afford 2, 6-dimethoxyphenol and 2-methoxyphenol in sequentially and rapidly ascended yields, while the phenylpropanol analogues were not detected during the reaction, which evidenced that the deconstruction of the $C_{sp2}-C_{sp3}$ bond was occurred locally on the molecular structure of the reactant and accompanied by the HY$_{30}$ catalyzed hydrolysis of the aliphatic carbon–oxygen ($C_\beta-O$, Fig. 1c) bonds[56]. At this stage, water was the only reactant besides **11a** in the formation of the phenolic hydroxyl (–OH) groups, and no active hydrogen was needed. With the ongoing refining process, the methoxyl derived $C_{sp2}-O(CH_3)$ bonds in the above monomeric intermediates were gradually transformed into the $C_{sp2}-H$ bonds. Meanwhile, the phenolic $C_{sp2}-O$ bonds with higher bond energy could also be cleaved along with the $C_{sp2}-O(CH_3)$ bond, which should be attributed to the RuW catalyzed hydrogenolysis of the phenolic hydroxyl group using the active hydrogen derived from the formaldehyde molecules generated during the SSH reaction of the $C_{sp2}-O(CH_3)$ bond (Supplementary Fig. 2). To explore the mechanism of dehydroxylation, we conducted the reaction of 4-(1-hydroxypropyl)phenol over the RuW/HY$_{30}$ catalyst in the formaldehyde solution (Supplementary Table 8), where thermolabile paraformaldehyde was used as the source of formaldehyde to simulate the gradual supply of the formaldehyde generated from the SSH reaction. As expected, only the $C_{sp2}-C_{sp3}$ bond in 4-(1-hydroxypropyl)phenol was deconstructed over the HY$_{30}$ component with phenol as the sole product, and the cleavage of the phenolic $C_{sp2}-O$ bond did not occur without the source of formaldehyde (Supplementary Table 8, entry 1). With the introduction of formaldehyde, the benzene product was detected (Supplementary Table 8, entry 2), suggesting that the phenolic $C_{sp2}-O$ bond could be hydrogenolyzed over the RuW component using the active hydrogen derived from formaldehyde. Moreover, the yield of benzene was steadily increased with the increase of the source of formaldehyde (Supplementary Table 8, entries 2–8), which confirmed that the phenolic $C_{sp2}-O$ bonds can be efficiently hydrogenolyzed over the RuW component with the active hydrogen derived from the gradually increased formaldehyde molecules during the RuW catalyzed SSH reaction of the $C_{sp2}-O(CH_3)$ bonds.

**In situ refining of lignins**. We ultimately moved to the refining of the real lignins that extracted from a variety of woods, including pine, cedrela, poplar, willow, eucalyptus, peach, applewood and cedar, and herbaceous plant Phyllostachys pubescens. As illustrated in Fig. 5 and Supplementary Table 9, benzene could be effectively abstracted from the above lignins in different yields under the function of this in situ refining strategy. For comparison, a blank experiment (without catalyst) was also performed using the pine lignin under the same conditions, which could not yield any low-molecular weight products (Supplementary Fig. 12), suggesting the crucial catalysis of the employed refining system. Remarkably, the in situ refining system operated transformation of the lignin is selective, and the pine lignin, for example, could be exclusively refined into benzene product with a maximum yield of 18.8% (based on lignin, Fig. 5, Supplementary Fig. 13 and Supplementary Table 9). In addition to the benzene product, some intermediate products, for example 2, 6-dimethoxyphenol, 2-methoxyphenol and phenol, could be detected during the reaction (Supplementary Fig. 14). As the reaction proceeded, the above intermediates were further transformed with benzene as the only liquid product (Supplementary Fig. 13), coinciding with the course of the reaction of **11a** (Supplementary Fig. 11). These overwhelming evidences point out that the HY$_{30}$ and RuW centers respectively catalyzed the reactions of the $C_{sp2}-C_{sp3}$ and $C_{sp2}-O$ bonds in sequence, and their cooperation worked effactually on the refining of the H (p-hydroxyphenyl), G (guaiacyl) and S (syringyl)-derived phenylpropanol building blocks in lignin [at 130.57/7.67, 110.64/6.94 and 104.00/6.72 ppm, colored with green, magenta and blue] (Supplementary Fig. 15)[57,58]. After the reaction, the $^1H/^{13}C$ 2D heteronuclear single-quantum coherence (HSQC) and quantitative $^{13}C$ NMR analyses of the residue lignin oil showed a significant decrease in the signals of the above mentioned H, G, and S-derived phenylpropanol structures, which was accompanied by a simultaneous decrease in the quantity of the methoxy group [at 55.15/3.75 ppm, colored with wine] (Supplementary Fig. 16)[57,58], indicating that the phenylpropanol structures were consumed and converted fruitfully into benzene product with methoxy-supplied hydrogen source in the refining system. Furthermore, the in situ refining protocol could be readily generalized toward a range of lignins with different content of the phenylpropanol structures and methoxy group/benzene ring ratios (Supplementary Figs. 15, 17–24), and their respective yields of benzene product were all above 10% (based on lignin, Fig. 5 and Supplementary Table 9), establishing lignin as a feasible resource for benzene production. Notably, the yields of benzene product were not always proportional to the content of phenylpropanol structure (Supplementary Table 9), which is related to the contents of the S, G, and H units in lignin. Specifically, the mass yield of benzene abstracted from S units is sequentially lower than those from the equivalent G and H units. In contrast, pine lignin has more G and H units (Supplementary Fig. 15b), but eucalyptus lignin contains more S units (Supplementary Fig. 20b), which leads to a lower mass yield of benzene obtained from eucalyptus lignin, albeit with larger content of the phenylpropanol structures (Supplementary Table 9). To get pure benzene, we conducted a scale-up experiment for the transformation of the pine lignin, which produced 8.5 g of pure benzene from 50.0 g of lignin (Supplementary Fig. 25). The yield of benzene was slightly reduced comparing with the normal scale experiment (Supplementary Fig. 13), mainly owing to the loss of benzene in the transfer and separation processes. Based on the experimental results, we know that the lignin-to-benzene route integrates two steps, including lignin extraction and catalytic valorization of lignin, which can not only preserve the native structure of lignin for better understanding of the genuine reactivity of lignin, but more importantly, can free the lignin

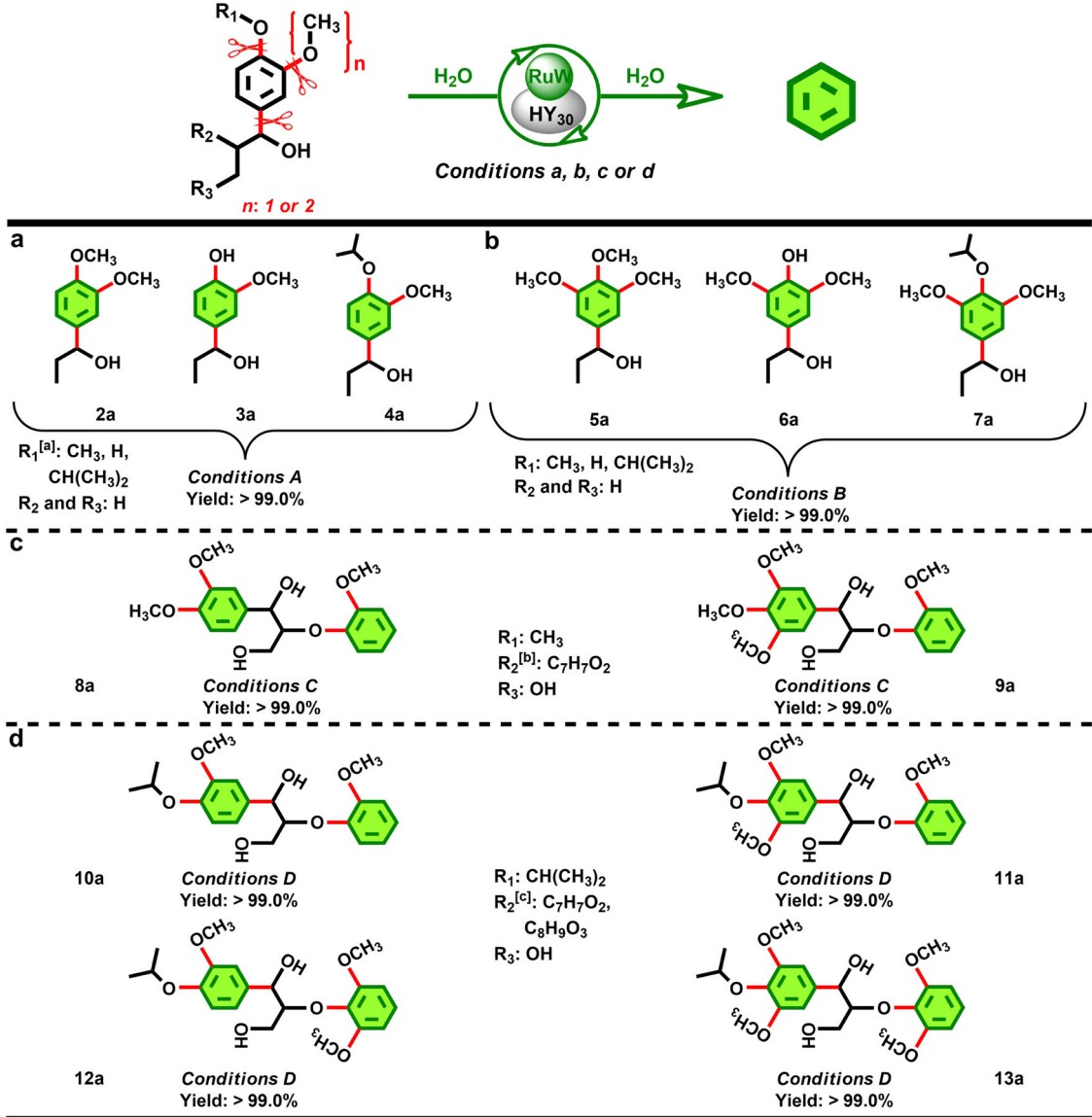

**Fig. 4 In situ refining strategy: portfolio scope of the $C_{sp2}$–$C_{sp3}$ and $C_{sp2}$–O bonds.** The general reaction scheme is shown at the top. The chemical bonds colored with red is transformed, and the benzene ring colored with green is the desired product from the in situ refining strategy. Yields of benzene product provided are at full conversion of substrates, as averages of three experiments conducted in parallel. [a]$CH(CH_3)_2$, isopropyl. [b]$C_7H_7O_2$, 2-methoxyphenoxyl. [c]$C_8H_9O_3$, 2, 6-dimethoxyphenoxyl. Reaction conditions: $H_2O$ (5.0 mL), 0.1 MPa $N_2$, 800 rpm. **Conditions a:** Substrate (1.0 mmol), RuW/HY$_{30}$ catalyst (0.25 g, 3.0 wt% Ru, 17 wt% W), 190 °C, 7.0 h. **Conditions b:** Substrate (1.0 mmol), RuW/HY$_{30}$ catalyst (0.30 g, 3.0 wt% Ru, 17 wt% W), 200 °C, 7.5 h. **Conditions c:** Substrate (0.5 mmol), RuW/HY$_{30}$ catalyst (0.35 g, 3.5 wt% Ru, 20 wt% W), 210 °C, 8.0 h. **Conditions d:** Substrate (0.5 mmol), RuW/HY$_{30}$ catalyst (0.35 g, 3.5 wt% Ru, 20 wt% W), 210 °C, 8.0 h.

transformation from the interference of the reaction of the carbohydrate in wood powder. In the first step, lignin was extracted from wood powder by solid–liquid separation and solvent recuperation, during which the used wood powder was also recovered along with the organic solvent and then used in the continual extraction process. In the second step, the extracted lignin was fed to the catalytic reactor only with water, and exclusively converted into benzene product, where the catalyst could be recovered and reused. Meanwhile, as the only liquid product, benzene could be quite easily separated from the system without complex procedures. The sufficiently recyclable and highly selective features of the above processes are beneficial to producing benzene economically. From the perspective of atomic economy, the active hydrogen atoms in the lignin molecule could also be utilized successfully along with the abstraction of the

benzene rings from lignin under the in situ refining strategy. Moreover, the lignin residue obtained in the lignin conversion process can be collected and further valorized into high value-added fuel products and chemicals. Given the above advantageous features, this lignin-to-benzene route has the potential of industrial application.

## Discussion

In summary, we have developed an in situ refining strategy to the transformation of the $C_{sp2}$–$C_{sp3}$ and $C_{sp2}$–O bonds for the sustainable production of benzene from lignin using RuW/HY$_{30}$ as the multifunctional catalyst and water as the reaction medium, which further draws attention to biomass valorization methods. It demonstrates that the HY$_{30}$ zeolite and RuW components in the

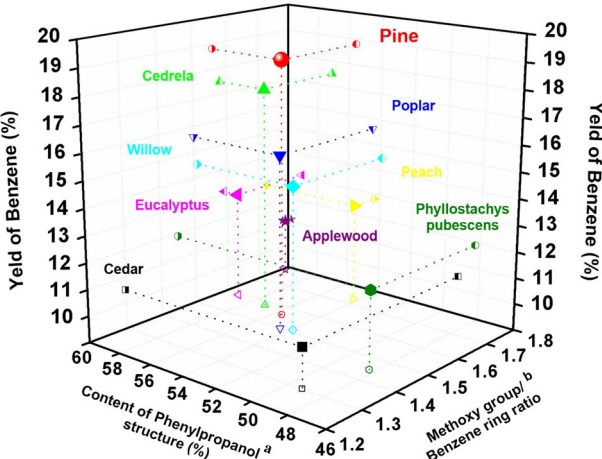

**Fig. 5 In situ refining of lignins.** 3D plot for the comparison of benzene yield-content of phenylpropanol structure-methoxy group/benzene ring ratio of the employed lignins. Full data are listed in Supplementary Table 9. Yields of benzene product provided are the averages of three experiments conducted in parallel. [a]Phenylpropanol structure, $[-(H_3CO)_nPh-C_\alpha H(OH)-$, $n = 0$, 1 and 2], the content of the defined phenylpropanol structure in lignin is calculated on the basis of the quantitative analysis (Supplementary Figs. 15, 17–24) of the $C_\alpha$ atom whose quantity is equal to that of the phenylpropanol structure. The formula is displayed in Supplementary Table 9. [b]Methoxy group/benzene ring ratio, the ratio of the quantity of methoxy group substituted on the benzene ring to the quantity of benzene ring in lignin. The formula is displayed in Supplementary Table 9. Reaction conditions: lignin (0.50 g), RuW/HY$_{30}$ (0.50 g, 3.5 wt% Ru, 20 wt% W), $H_2O$ (6.5 ml), 240 °C, 12 h, 0.1 MPa $N_2$, 800 rpm.

RuW/HY$_{30}$ catalyst act in synergy to orderly deconstruct the $C_{sp2}$–$C_{sp3}$ and $C_{sp2}$–O bonds in lignin structure, by which benzene can be exclusively produced from lignin with a maximum benzene yield of 18.8% on the lignin weight basis, highlighting the significance of developing catalytic technologies for the aromatic building blocks in lignin. Innovatively, the RuW component can not only catalyze the hydrogenolysis of the $C_{sp2}$–O bond using the active hydrogen in situ abstracted from lignin molecule, but more importantly, allow Bronsted acid sites of HY$_{30}$ zeolite to promptly deconstruct the $C_{sp}$–$C_{sp3}$ bonds on the local structure of lignin molecule without any precedent reductive catalytic fractionation process and competition from the hydrogenolysis of the hydroxyl group in $[C_{sp2}$–$C_{sp3}(OH)]$ motif. In the scale-up experiment, 8.5 g of benzene can be produced from 50.0 g of lignin without any saturation byproducts. This in situ lignin refining strategy liberates the trapped benzene rings from the molecular structure of lignin, and paves a new way for sustainable production of benzene using lignin as the feedstock, which has great potential of practical application.

## Methods

**Catalyst preparation.** Supported RuW/Zeolite catalysts were prepared by commonly used wet impregnation method[43]. Prior to the impregnation, the used zeolites were pretreated by calcination at 550 °C for 4 h. In a typical preparation, ammonium metatungstate (AMT) (1.25 g) and RuCl$_3$·xH$_2$O (0.40 g) were dissolved in 15 mL of deionized water, respectively. Then, the precursor solutions containing AMT and Ru$^{3+}$ ion were successively added dropwise to 70 mL deionized water with 2.0 g of zeolite at room temperature. The obtained mixture was vigorously stirred for 48 h, evaporated, and dried at 473 K for 16 h in an oven. The as-prepared sample was reduced in a continuous 10% H$_2$/Ar flow, from room temperature to 400 °C at 5 °C min$^{-1}$ and then to 900 °C at 1 °C min$^{-1}$, and maintained at 900 °C for 1 h. After being cooled to room temperature under Ar atmosphere, the reduced catalyst was exposed to 1% O$_2$/Ar atmosphere for 1 h to form a passivation layer to prohibit against bulk oxidation before exposure to air. The passivated catalyst was kept under an inert atmosphere before testing and characterizations, and denoted as RuW/Zeolite$_x$ (x, the framework Si/Al ratio of the

zeolite). For comparison, supported Ru/Zeolite$_x$, W/Zeolite$_x$ and other catalysts were prepared by the same procedures for preparing the RuW/Zeolite$_x$ catalyst.

**Catalyst characterization.** N$_2$ adsorption-desorption of the samples were measured using a Micromeritics Tristar II 3020 at liquid nitrogen temperature. The specific surface areas were calculated by using the Brunauer–Emmett–Teller model. The pore size distribution of the sample was calculated using the Barret–Joyner–Halenda pore size model. XRD measurements were conducted on an X-ray diffractometer (D/MAX-RC, Japan) operated at 40 kV and 200 mA with Cu Kα (λ = 0.154 nm) radiation. Rietveld refinements were performed applying the TOPAS program[59] for the measured XRD profile of the RuW/HY$_{30}$ catalyst. The XPS measurements were carried out on an ESCAL Lab 220i-XL spectrometer at a pressure of ~3 × 10$^{-9}$ mbar (1 mbar = 100 Pa) using Al Kα as the excitation source ($hv$ = 1486.6 eV) and operated at 15 kV and 20 mA. The X-ray absorption find structure (XAFS) spectra (Ru K-edge and W L$_3$-edge) were collected at 1W1B station in Beijing Synchrotron Radiation Facility (BSRF). The storage ring of BSRF was operated at 2.5 GeV with a maximum current of 250 mA. Using Si (111) double-crystal monochromator, the data collection were carried out in transmission mode using ionization chamber. All spectra were collected in ambient conditions. The acquired EXAFS data were processed according to the standard procedures using the ATHENA module implemented in the IFEFFIT software packages. The k$^2$-weighted EXAFS spectra were obtained by subtracting the post-edge background from the overall absorption and then normalizing with respect to the edge-jump step. Subsequently, k$^2$-weighted χ(k) data of Ru K-edge and W L$_3$ edge were Fourier transformed to real (R) space using a han-ning windows (dk = 1.0 Å$^{-1}$) to separate the EXAFS contributions from different coordination shells. To obtain the quantitative structural parameters around central atoms, least-squares curve parameter fitting was performed using the ARTEMIS module of IFEFFIT software packages[60]. TEM and HRTEM images were obtained on a JEOL-2011F electron microscope operating at 200 kV. The contents of supported metals on the catalysts were determined by ICP.

**Lignin extraction and characterization.** In a typical experiment, 70 g of wood powder, 490 mL of acetone and 210 mL of water were loaded into a 1-L autoclave (Weihai Xinyuan Chemical Machinery Co. Ltd.). The autoclave was sealed and purged with N$_2$ to remove the air at room temperature and subsequently charged with 0.1 MPa of N$_2$. Then, the autoclave was heated to 160 °C within 1 h. After that, the stirrer was started with a stirring speed of 600 rpm, and the reaction time was recorded. After 1 h, the autoclave was cooled down quickly, and the gas was released. The liquid was collected by a filtration process. The filter residue was washed three times with 200 mL of acetone/H$_2$O (9:1), and the washing liquor was collected and combined with the filtrate liquid. After that, the liquid was concentrated under vacuum at 30 °C until the liquid became muddy and then dissolved again by 30 mL of acetone. Then, the concentrated lignin solution was slowly poured into the rapidly stirred 2000 mL of water, and the precipitate was filtered using a funnel with a pore size of 3–4 μm. Finally, the lignin solid was freeze-dried under vacuum for 24 h, and the extracted lignin was obtained. In addition, the solvent used in the above procedures were completely recovered along with the used wood powder (filter residue), and reused in the next extraction experiment. The obtained lignin was dissolved in 550 μl of dimethyl sulfoxide (DMSO)–d6 and characterized by $^1$H/$^{13}$C 2D HSQC NMR and quantitative $^{13}$C NMR analyses using Bruker Avance III 500WB and Bruker Avance 600, as described by other researchers[57].

**Catalytic performance.** The reaction was carried out in a Teflon-lined stainless-steel reactor of 20 mL with a magnetic stirrer. In a typical experiment, a suitable amount of reactant, catalyst, and water were loaded into the reactor. The reactor was sealed and purged with N$_2$ for three times to remove the air at room temperature and subsequently charged with desired gas. Then the reactor was placed in a furnace at desired reaction temperature. When the reactor reached the desired reaction temperature, the stirrer was started with a stirring speed of 800 rpm, and the reaction time was recorded. After the reaction, the reactor was placed in ice water, and the gas was released, passing through the ethyl acetate. The reaction mixture in the reactor was transferred into a centrifuge tube. Then the reactor was washed with the ethyl acetate used for the gas filtration, which was finally combined with the reaction mixture. After centrifugation, the catalyst was separated from the reaction mixture. The quantitative analysis of the liquid products in the organic phase was conducted using a GC (Agilent 6820) equipped with a flame ionization detector and HP-5MS/HP-INNOWAX capillary columns (0.25 mm in diameter, 30 m in length). Identification of the products and reactant was performed using a GC-MS [Agilent 5977A, HP-5MS capillary column (0.25 mm in diameter, 30 m in length)] and by comparing the retention time to respective standards in GC traces. Biphenyl was used as the internal standard to determine the conversions of substrates, selectivities and yields of the products. Identification of the products in the aqueous phase was conducted by $^1$H NMR analysis on a Bruker Avance III 400 HD with D$_2$O as the solvent. The carbon balance for the reaction of the model compounds was calculated using C$_{aromatics}$ balance which was given relative to the aromatic products[43]. The C$_{aromatics}$ balances for the reaction of the model compounds were better than 99%. The recovered lignin residual solid

were analyzed by $^1H$/$^{13}C$ 2D HSQC NMR and quantitative $^{13}C$ NMR spectroscopy (Bruker Avance III 600 HD).

**Recycling of the catalyst**. The reusability of RuW/HY$_{30}$ catalyst was tested using the reaction of the model compound 1-(4-methoxyphenyl)-1-propanol (**1a**). After the reaction, the reaction mixture in the reactor was transferred into a centrifuge tube. Then the reactor was washed with ethyl acetate, which was combined with the reaction mixture. Subsequently, the reaction mixture was centrifuged and the ethyl acetate layer was analyzed by GC. After that, the used RuW/HY$_{30}$ catalyst was separated from the reaction mixture and successively washed with ethanol ($5 \times 10$ mL) and water ($5 \times 10$ ml). Then, the recovered catalyst was reused directly for the next run.

**Detection of intermediates**. The detection of intermediates was carried out in a Teflon-lined stainless-steel reactor of 20 mL with a magnetic stirrer. In the experiment, 1-(4-methoxyphenyl)-1-propanol (**1a**) (1 mmol), RuW/HY$_{30}$ (0.40 g) and H$_2$O (5.0 ml) were loaded into the reactor. The reactor was sealed and purged with N$_2$ three times to remove the air at room temperature and subsequently charged with 0.1 MPa of N$_2$. When the reactor was placed in a furnace and heated to 180 °C, the stirrer was started with a stirring speed of 800 rpm, and the reaction time was recorded. After 10 min, the reactor was transferred to a bath of liquid nitrogen very quickly. When the reaction mixture was frozen, the gas was released immediately. The reaction mixture was freeze-dried under vacuum for 12 h to remove the water. The detection of the intermediates in the dried sample was conducted by $^{13}C$ NMR, solid-state 2D $^{13}C\{^1H\}$ dipolar-mediated HETCOR, $^{27}Al$ MQMAS and $^{27}Al$ NMR analyses on Bruker Avance III 400.

**Mass balance analysis of the lignin transformation**. After the reaction of lignin, the gas was released, passing through the ethyl acetate. Then, the reaction mixture in the reactor was transferred into a centrifuge tube. After that, the reactor was washed with the ethyl acetate used for the gas filtration, which was finally combined with the reaction mixture. By centrifugation, the solid was separated from the reaction mixture, and the yield of the detectable products in the ethyl acetate layer was determined by GC. The separated solid was successively washed with acetone, and the used catalyst was recovered. Then, the collected liquid was subjected to rotavap to remove acetone solvent, and the lignin residue was obtained. Finally, the lignin residue and recovered catalyst were freeze-dried under vacuum for 24 h. The mass of the recovered catalyst was nearly the same as that of the catalyst initially loaded. The mass balance for the transformation of lignin was $92 \pm 5\%$, which was calculated using Eq. (1)[56].

$$\text{Mass balance} = \frac{\text{Detectable products} + \text{residual lignin oil}}{\text{Lignin loaded}} \times 100\% \qquad (1)$$

**Scale-up transformation of the pine lignin**. The scale-up reaction was performed in a 1-L autoclave (Weihai Xinyuan Chemical Machinery Co. Ltd.). In the reaction, 50.0 g of lignin, 50.0 g of RuW/HY$_{30}$ catalyst and 650 mL of H$_2$O were loaded into the autoclave. The autoclave was sealed and purged with N$_2$ to remove the air at room temperature and subsequently charged with 0.1 MPa of N$_2$. Then, the autoclave was heated to 240 °C, and the autoclave reached the desired reaction temperature within 60 min. After that, the stirrer was started with a stirring speed of 800 rpm, and the reaction time was recorded. After the reaction, the autoclave was cooled to room temperature, and the gas was released. The liquid layers were transferred into a separatory funnel, and then the aqueous layer was removed. Desired benzene product was finally obtained. The identification of the benzene product was conducted using a GC-MS [Agilent 5977A, HP-5MS capillary column (0.25 mm in diameter, 30 m in length)] and by comparing the retention time to standard benzene in GC. $^1H$ and $^{13}C$ NMR analyses (Bruker Avance III 400) were also performed to detect the benzene product in the organic layer.

## Data availability

All data needed to evaluate the conclusions in the paper are present in the paper and/or the Supplementary Information. Additional data available from authors upon request.

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

## Acknowledgements

We thank Qian (Q.) Li for his assistance in the NMR measurements. The work was supported financially by National Natural Science Foundation of China (22073105), National Key Research and Development Program of China (2017YFA0403103), Beijing Municipal Science & Technology Commission (Z191100007219009).

## Author contributions

Q.M. and B.H. conceived and designed the present work. Q.M. and B.H. wrote the manuscript. Q.M. and J.Y. conducted all of the experimental work. R.W., H.L., Y.S., N. W., J.X., L.Z., and J.Z. assisted with SEM, TEM, XRD, NMR, and XAFS measurements. All authors discussed the results and contributed to the final manuscript.

## Competing interests

The authors declare no competing interests.
