## [Peer Review File · Nature Communications]

REVIEWER COMMENTS

Reviewer #1 (Remarks to the Author):

The manuscript discusses the use of bifunctional RuW/HY catalysts for the selective conversion of lignin derivatives to benzene. The catalyst and reaction system are certainly interesting. However, there are several significant concerns:

1. The authors claim that it is "unprecedented" to have a hydrodeoxygenation catalyst that removes oxygen from lignin derivatives without some hydrogenation of the aromatic ring. In fact, there are numerous manuscripts discussing MoO₃ catalysts that do exactly that, e.g., Prasomsri, et al., *Energy Environ. Sci.*, 2014, 7, 2660-2669; Saraeian, et al., *Green Chem.*, 2020, 22, 2513-2525. Although it should be noted that low pressure H₂ is needed in those reaction systems.
2. The second significant claim is the reaction system used in this work can perform the desired transformation without the addition of H₂ by "using the abstracted hydrogen from the methoxyl group". I do not understand how the overall hydrogen stoichiometry works for the proposed reaction. If 1-(4-methoxyphenyl)-1-propanol forms benzene, methanol and isopropanol, 2 moles of added H₂ are required. The H₂ requirement goes up even more if species 1e and 1g (Table 1) are formed. Where is that excess hydrogen coming from?

Reviewer #2 (Remarks to the Author):

The work reported by Meng, Han, and coworkers presents very interesting and promising results that can draw significant attention. This work has the broad impact to appeal to your community-spanning readership.

In lignin valorization, the diverse range of phenolic compounds produced after lignin depolymerization is often discussed as a technical challenge. Thus, developing strategic depolymerization processes that can yield products with high selectivity and yield will be highly desirable.

Also, considering that the lignin-to-benzene approach is not readily realized due to the diverse functionality with different bond dissociation enthalpy, the proposed catalytic process can be a key strategy in lignin valorization.

However, there are several points the authors should address to meet the rigorous standards of Nat Comm.

1) Novelty of the catalyst

Ruthenium-tungsten bifunctional catalyst has been implemented in the lignin depolymerization (<https://pubs.rsc.org/en/content/articlehtml/2015/gc/c5gc00326a>), and extensively studied in the previous publication from the same group (10.1126/sciadv.aax6839). Somehow it seems the present

work simply followed the previous work, despite some new findings of Csp2-Csp3 cleavage with the effect of supporting materials.

2) Dehydroxylation

One missing point of the current work is that dehydroxylation is not significantly explored. Considering that phenolic hydroxyl group is the most frequent functionality existing in lignin structure and dehydroxylation is required to produce benzene from lignin, this reviewer believes that in-depth mechanistic insights of dehydroxylation should be explored (this is a different Csp2-O when compared with that in methoxyl groups). In addition, bond dissociation energy of C-O in Ar-OH is higher than that in Ar-OCH₃, it should be considered when developing reaction pathways / mechanisms. Although the authors tested many different model compounds that contain the phenolic hydroxyl groups and showed the Ar-OH bond can be readily broken (in Fig. 4), it would be more comprehensive if the dehydroxylation is included.

3) In situ refining

When the hydrogen gas is introduced, the selectivity of benzene is lowered and yields of other saturated compounds and alkylbenzene increased. There seem to be competing reactions between in-situ hydroprocessing and hydrogenolysis by external hydrogen gas.

Also, the SiO₂ / Al₂O₃ ratio did affect the yield and selectivity of benzene without the external hydrogen gas, there are multiple factors that determines the reaction and product distributions.

Please discuss this more comprehensively regarding this matter.

Different reaction conditions were used to test several model compounds (Fig. 4). Why? What rationale?

With the real lignin, more severe reaction conditions were used.

4) What is the main role of water?

5) Lignin-to-benzene

The results indicate that the yield of benzene is not proportional to the content of phenylpropanol structure. What would be the main determining factor?

Mass balance analysis is necessary. The authors only discuss the production of benzene. To evaluate the overall process efficiently, the mass balance should be presented.

6) TEA

Please consider including technoeconomic insights of the proposed process.

Reviewer #3 (Remarks to the Author):

For future biorefineries it is of great economical importance to obtain value added products from the lignin part of lignocellulosic biomass. One such product could be benzene, which is currently produced from fossil resources. Previous efforts in this area have not been successful in producing significant

quantities of benzene from processing of either biomass directly or from isolated lignin. In a previous contribution by the authors (ref 31 of the paper), they showed that a similar catalyst (RuW/SiO₂) was able to produce arenes from lignin. In this paper, it is reported that high-silica HY zeolite supported RuW alloy catalyst enables in situ refining of lignin to produce benzene in significant yields – a maximum of 18.8 % on lignin weight basis. For several lignin-type model compounds, the yield of benzene is almost 100 %. The experimental results are highly interesting and of great novelty and represent a significant step forward compared to previous papers in refining lignin to value added products. The authors show that the catalysis occurs by Bronsted acid catalyzed transformation of the Csp²-Csp³ bonds on the local structure of lignin molecule and RuW catalyzed hydrogenolysis of the Csp²-O bonds using hydrogen abstracted in-situ from the lignin molecule. Interestingly, the chemistry takes place using water as solvent. The reaction mechanism is elucidated in detail by a combination of control experiments and density functional theory calculations. The authors also (modestly) up-scale their experiments to produce 8.5 g of benzene from 50 g of lignin.

In the proposed process, lignin is first extracted from the biomass by a solvent and then processed. It would be very relevant if the authors could comment on the economy of this two-step process. What should the benzene yield be for the process to be cost neutral ?

Please state the mass of catalyst applied in the experiments in table 1. It seems it is also not stated in the extended data file. This is necessary.

The authors use expensive Ru in their catalysts. This may likely impede the industrial implementation of the discovered chemistry. Did the authors test any non-noble metals as substitute for Ru ?

Why was the experiments with lignin operated at 240 C, when the model compound experiments were done at 180 C ?

It would be useful if the authors could comment on the other products formed from the processing real lignin. Were any other useful products than benzene formed ?

Overall the present work has high novelty and provides sufficient detail to allow reproduction with the possible exceptions noted above. I recommend publication.

Reviewer #1 (Remarks to the Author):

The manuscript discusses the use of bifunctional RuW/HY catalysts for the selective conversion of lignin derivatives to benzene. The catalyst and reaction system are certainly interesting. However, there are several significant concerns:

Response: We thank the reviewer for the positive comment very much.

1. The authors claim that it is "unprecedented" to have a hydrodeoxygenation catalyst that removes oxygen from lignin derivatives without some hydrogenation of the aromatic ring. In fact, there are numerous manuscripts discussing MoO₃ catalysts that do exactly that, e.g., Prasomsri, et al., *Energy Environ. Sci.*, 2014, 7, 2660-2669; Saraeian, et al., *Green Chem.*, 2020, 22, 2513-2525. Although it should be noted that low pressure H₂ is needed in those reaction systems.

Response: We thank the reviewer for the comment very much. The references [Saraeian, A. et al. *Green Chem.* **22**, 2513-2525 (2020); Prasomsri, T. et al. *Energy Environ. Sci.* **7**, 2660-2669 (2014);] have been cited as reference 34 and 46 in our revised manuscript. (Please see Page 21, marked in red).

In the original manuscript, we claimed that "... the 'unprecedented' efficiency of the in situ refining strategy for the benzene production is attributed to the excellent cooperation of RuW NPs and HY₃₀ zeolite in the catalyst, which catalyzed C_{sp2}-O SSH and C_{sp2}-C_{sp3} transformation...". This means that the production of benzene from lignin requires both hydrogenolysis of the C_{sp2}-O bonds and deconstruction of the C_{sp2}-C_{sp3} bonds under the functions of RuW NPs and HY₃₀ zeolite. Although the hydrodeoxygenation processes can also be achieved to remove oxygen from lignin derivatives without hydrogenation of the aromatic ring, using other hydrodeoxygenation catalysts, such as MoO₃, under low pressure of H₂ (*Energy Environ. Sci.*, 2014, 7, 2660-2669; *Green Chem.*, 2020, 22, 2513-2525), such arene products mainly consist of alkylbenzenes when the benzene ring of the reactants are substituted by the alkyl groups (*Green Chem.*, 2020, 22, 2513-2525), demonstrating that the C_{sp2}-C_{sp3} bonds are unbroken during the reaction. To support the advanced functionality of the in situ refining strategy, we have also conducted the reaction of 1-(4-methoxyphenyl)-1-propanol (**1a**) over the RuW/HY₃₀ catalyst under low pressure of exogenous H₂ (0.1 MPa), and the result has been listed in Table 1 in the revised manuscript as entry 6. Though the selectivities of the saturation by-products, such as cyclohexane and n-propylcyclohexane, decreases obviously in comparison to that under high pressure of exogenous H₂ (1.0 MPa, Table 1, entry 5), RuW catalyzed hydrogenolysis of the hydroxyl group substituted at the aliphatic C_{sp3} position [C_{sp2}-C_{sp3}(OH)] still occurs in the presence of exogenous H₂ and generates the n-propylbenzene product that cannot be further converted into benzene under the mild condition in this work (Table 1, entry 6), lowering the selectivity of benzene product.

Taken together, it is the innovative cooperation of RuW NPs and HY₃₀ zeolite that affords an extraordinary efficiency for the single production of benzene from lignin, where RuW NPs catalyzed self-supported hydrogenolysis (SSH) of the C_{sp2}-O bonds with the in situ abstracted hydrogen from the reactant can not only completely avoid

the saturation of the benzene ring, but more importantly, allow the $C_{sp^2}-C_{sp^3}$ bonds to be deconstructed promptly over the HY₃₀ zeolite without the competition from the hydrogenolysis of the hydroxyl group in [$C_{sp^2}-C_{sp^3}(OH)$] motif.

In order to disambiguate the description, we have reorganized this in the revised manuscript by “On the basis of the above comparison tests (Table 1), and given that traditional hydroprocessing with exogenous hydrogen source resulted not only in the unavoidable hydrogenation of the benzene ring to cyclohexane but also in the n-propylbenzene byproduct that could not be further converted into benzene under the mild condition (Table 1, entries 5 and 6), we proposed that the outstanding efficiency of the refining strategy for benzene should be attributed to the innovative matching of RuW NPs and HY₃₀ zeolite in the catalyst, where RuW NPs catalyzed hydrogenolysis of the $C_{sp^2}-O$ bonds with the in situ abstracted hydrogen from the reactant can not only completely avoid the saturation of the benzene ring, but more importantly, allow the $C_{sp^2}-C_{sp^3}$ bonds to deconstruct promptly over the HY₃₀ zeolite without the competition from the hydrogenolysis of the hydroxyl group in [$C_{sp^2}-C_{sp^3}(OH)$] motif.” (Please see Page 9, marked in red)

2. The second significant claim is the reaction system used in this work can perform the desired transformation without the addition of H₂ by "using the abstracted hydrogen from the methoxyl group". I do not understand how the overall hydrogen stoichiometry works for the proposed reaction. If 1-(4-methoxyphenyl)-1-propanol forms benzene, methanol and isopropanol, 2 moles of added H₂ are required. The H₂ requirement goes up even more if species 1e and 1g (Table 1) are formed. Where is that excess hydrogen coming from?

Response: We thank the reviewer for the comment very much. In this work, conversion of 1-(4-methoxyphenyl)-1-propanol (**1a**) to benzene product requires both hydrogenolysis of the $C_{sp^2}-O$ bonds and deconstruction of $C_{sp^2}-C_{sp^3}$ bonds under the functions of RuW NPs and HY₃₀ zeolite, where RuW NPs catalyzes self-supported hydrogenolysis (SSH) of $C_{sp^2}-O$ bond using the in situ abstracted hydrogen from the reactant (Supplementary Fig. 2), and HY₃₀ zeolite deconstructs the $C_{sp^2}-C_{sp^3}$ bonds via a stepwise protonated dehydroxylation, γ -methyl shift and $C_{sp^2}-C_{sp^3}$ β scission pathway (Fig. 3). Specifically, the RuW catalyzed SSH reaction of the $C_{sp^2}-O$ bond in **1a** requires 1 mole of the active H atoms abstracted from the methoxy group, as shown in Supplementary Fig. 2, and meanwhile, the methoxy group is converted into formaldehyde molecule (not methanol). The deconstruction of the $C_{sp^2}-C_{sp^3}$ bonds in **1a** over the HY₃₀ zeolite does not require active H atoms, as shown in Fig.3a. Overall, the methoxy groups can supply sufficient active H atoms for the conversion of 1-(4-methoxyphenyl)-1-propanol (**1a**) into benzene product.

However, when SiO₂ (Table 1, entry 9), Al₂O₃ (Table 1, entry 10), HY zeolites with lower framework Si/Al ratios (HY₃, HY₅ and HY₁₅, Table 1, entries 11-13), Beta zeolite (Table 1, entry 15), HZSM-5 zeolites (Table 1, entries 16-21), Mordenite zeolite (Table 1, entry 22), SAPO zeolite (Table 1, entry 23) and MCM-41 zeolite (Table 1, entry 24) were employed as the support in our multifunctional catalyst, their

selectivities to benzene product were much lower than those over RuW/HY₃₀ catalyst, whereas n-propylbenzene (**1e**) was generated as the main product. This is due to the poor activities of the above materials on the deconstruction of the C_{sp2}-C_{sp3} bonds that offered RuW the opportunity to hydrogenolyze the hydroxyl group substituted at the aliphatic C_{sp3} position of the [C_{sp2}-C_{sp3}(OH)] motif using the active hydrogen derived from the formaldehyde molecules generated during the SSH reaction of the C_{sp2}-O bonds. Similarly, the 1-methoxy-4-propylbenzene (**1g**) product was also resulted from the above hydrogenolysis of the hydroxyl group in [C_{sp2}-C_{sp3}(OH)] motif of **1a**, but without SSH reaction of the C_{sp2}-O bond proceeding yet.

To support the results in Table 1, we have supplemented the reaction of 1-phenylpropan-1-ol in formaldehyde solution, using RuW/SiO₂ (Table 1, entry 9), RuW/Al₂O₃ (Table 1, entry 10), RuW/HY (HY₃, HY₅ and HY₁₅, Table 1, entries 11-13), RuW/Beta (Table 1, entry 15), RuW/HZSM-5 (Table 1, entries 16-21), RuW/Mordenite (Table 1, entry 22), RuW/SAPO (Table 1, entry 23) and RuW/MCM-41 (Table 1, entry 24) as the catalysts, and the results have been listed in Supplementary Table 5. In order to simulate the gradual supply of the formaldehyde generated from the SSH reaction, thermolabile paraformaldehyde was used as the source of formaldehyde in the solution. It can be found that the 1-phenylpropan-1-ol was mainly transformed into the n-propylbenzene product under the same conditions in Table 1, proving that the RuW component could catalyze the hydrogenolysis of the hydroxyl group in [C_{sp2}-C_{sp3}(OH)] motif using formaldehyde as the hydrogen source. Thus, taking into account the chemical bonding environment of the [C_{sp2}-C_{sp3}(OH)] motif, achieving our envisioned benzene production rested on the innovative matching of the HY₃₀ zeolite and RuW components that would act in synergy to orderly deconstruct the stubborn C_{sp2}-C_{sp3} and C_{sp2}-O bonds.

According to the comment, in the revised manuscript, we have discussed this by “As the comparison of the utility of the RuW/HY₃₀ catalyst, it is the poor activities of the supports in the above catalysts (Table 1, entries 9-13 and 15-24) on the deconstruction of the C_{sp2}-C_{sp3} bonds that offered RuW the opportunity to hydrogenolyze the hydroxyl group substituted at the aliphatic C_{sp3} position of the [C_{sp2}-C_{sp3}(OH)] motif using the active hydrogen derived from the formaldehyde molecules generated during the SSH reaction of the C_{sp2}-O bonds (Supplementary Fig. 2), consequently leading to the n-propylbenzene byproduct (Supplementary Table 5). Similarly, the 1-methoxy-4-propylbenzene (**1g**) was also resulted from the above hydrogenolysis of the hydroxyl group in [C_{sp2}-C_{sp3}(OH)] motif of **1a**, but without SSH reaction of the C_{sp2}-O bond proceeding yet.” (Please see Page 7, marked in red)

Reviewer #2 (Remarks to the Author):

The work reported by Meng, Han, and coworkers presents very interesting and promising results that can draw significant attention. This work has the broad impact to appeal to your community-spanning readership.

In lignin valorization, the diverse range of phenolic compounds produced after lignin depolymerization is often discussed as a technical challenge. Thus, developing strategic

depolymerization processes that can yield products with high selectivity and yield will be highly desirable.

Also, considering that the lignin-to-benzene approach is not readily realized due to the diverse functionality with different bond dissociation enthalpy, the proposed catalytic process can be a key strategy in lignin valorization.

However, there are several points the authors should address to meet the rigorous standards of Nat Comm.

Response: We thank the reviewer for the positive comment very much.

1. Novelty of the catalyst

Ruthenium-tungsten bifunctional catalyst has been implemented in the lignin depolymerization (<https://pubs.rsc.org/en/content/articlehtml/2015/gc/c5gc00326a>), and extensively studied in the previous publication from the same group (10.1126/sciadv.aax6839). Somehow it seems the present work simply followed the previous work, despite some new findings of C_{sp2}-C_{sp3} cleavage with the effect of supporting materials.

Response: We thank the reviewer for the comment very much. The reference [Huang, Y. B. et al. *Green Chem.* **17**, 3010-3017 (2015)] has been cited as reference 36 in our revised manuscript. (Please see Page 21, marked in red).

The novelty of the RuW/HY₃₀ catalyst can be summarized from three aspects, firstly, in terms of the molecular level, production of benzene from lignin requires both hydrogenolysis of the C_{sp2}-O bonds and deconstruction of the C_{sp2}-C_{sp3} bonds. However, in the currently reported methodologies, such as catalytic pyrolysis, hydrodeoxygenation and combined catalytic processing, benzene could only be detected in quite low yields in the complex mixture containing the phenolic hydroxyl, methoxyl and alkyl-substituted aromatic products with the unbroken C_{sp2}-C_{sp3}/C_{sp2}-O bonds. Although ruthenium-tungsten catalysts have been reported and adept in the hydrogenolysis of the C_{sp2}-O bonds during the depolymerization of lignin [*Green Chem.* **17**, 3010-3017 (2015); *Sci. Adv.* **5**, eaax6839 (2019)], the C_{sp2}-C_{sp3} bonds could not be deconstructed over the ruthenium-tungsten catalysts and alkylbenzenes (toluene, ethylbenzene and propylbenzene) were the main products. Hence, HY₃₀ zeolite plays a crucial role in the desired transformation of lignin into benzene product. Secondly, the aliphatic C_α position (namely the C_{sp3} carbon in the C_{sp2}-C_{sp3} bond, Fig 1C) on the side chain of benzene rings in the lignin structures, is substituted by the hydroxyl group, and consequently, the HY₃₀ zeolite catalyzed deconstruction [C_{sp2}-C_{sp3}(OH)] bond is faced with the competition of the RuW catalyzed hydrogenolysis of the hydroxyl group when the exogenous hydrogen is used as reductant, generating the n-propylbenzene byproduct that could not be further converted into benzene under the mild condition (Table 1, entries 5 and 6). Innovatively, the RuW component in the RuW/HY₃₀ catalyst could not only catalyze the hydrogenolysis of the C_{sp2}-O bond using the active hydrogen in situ abstracted from the methoxy group [namely self-supported hydrogenolysis (SSH), Supplementary Fig. 2], but more importantly, allow the C_{sp2}-C_{sp3} bonds to deconstruct promptly over the HY₃₀ zeolite without the competition from the hydrogenolysis of the hydroxyl group in [C_{sp2}-C_{sp3}(OH)] motif. Thirdly, as shown in

Table 1, compared with HY₃₀ zeolite, the supports in the RuW/SiO₂, RuW/Al₂O₃ and RuW/zeolite (Table 1, entries 9-13 and 15-24) catalysts, could not deconstruct the C_{sp2}-C_{sp3} bonds promptly and efficiently, which also resulted in the competitive hydrogenolysis of the hydroxyl group in [C_{sp2}-C_{sp3}(OH)] motif using the active hydrogen derived from the formaldehyde molecules generated during the SSH reaction of the C_{sp2}-O bonds and led to the n-propylbenzene byproduct.

In view of the above discussion, HY₃₀ zeolite and RuW NPs act in synergy to orderly deconstruct the C_{sp2}-C_{sp3} and C_{sp2}-O bonds in the lignin structure. The design of the RuW/HY₃₀ multifunctional catalyst is an innovative way for the efficient and orientable transformation of lignin into benzene product with the desired yield.

In the revised manuscript, we have discussed this in more detail by “On the basis of the above comparison tests (Table 1), and given that traditional hydroprocessing with exogenous hydrogen source resulted not only in the unavoidable hydrogenation of the benzene ring to cyclohexane but also in the n-propylbenzene byproduct that could not be further converted into benzene under the mild condition (Table 1, entries 5 and 6), we proposed that the outstanding efficiency of the refining strategy for benzene should be attributed to the innovative matching of RuW NPs and HY₃₀ zeolite in the catalyst, where RuW NPs catalyzed hydrogenolysis of the C_{sp2}-O bonds with the in situ abstracted hydrogen from the reactant not only can completely avoid the saturation of the benzene ring, but more importantly, allow the C_{sp2}-C_{sp3} bonds to deconstruct promptly over the HY₃₀ zeolite without the competition from the hydrogenolysis of the hydroxyl group in [C_{sp2}-C_{sp3}(OH)] motif.” (Please see Page 9, marked in red)

“It demonstrates that the HY₃₀ zeolite and RuW components in the RuW/HY₃₀ catalyst act in synergy to orderly deconstruct the C_{sp2}-C_{sp3} and C_{sp2}-O bonds in lignin structure, by which benzene can be exclusively produced from lignin with a maximum benzene yield of 18.8% on the lignin weight basis, highlighting the significance of developing catalytic technologies for the aromatic building blocks in lignin. Innovatively, the RuW component can not only catalyze the hydrogenolysis of the C_{sp2}-O bond using the active hydrogen in situ abstracted from lignin molecule, but more importantly, allow Bronsted acid sites of HY₃₀ zeolite to promptly deconstruct the C_{sp2}-C_{sp3} bonds on the local structure of lignin molecule without any precedent reductive catalytic fractionation process and competition from the hydrogenolysis of the hydroxyl group in [C_{sp2}-C_{sp3}(OH)] motif.” (Please see Page 17, marked in red)

2. Dehydroxylation

One missing point of the current work is that dehydroxylation is not significantly explored. Considering that phenolic hydroxyl group is the most frequent functionality existing in lignin structure and dehydroxylation is required to produce benzene from lignin, this reviewer believes that in-depth mechanistic insights of dehydroxylation should be explored (this is a different C_{sp2}-O when compared with that in methoxyl groups). In addition, bond dissociation energy of C-O in Ar-OH is higher than that in Ar-OCH₃, it should be considered when developing reaction pathways/mechanisms. Although the authors tested many different model compounds that contain the phenolic hydroxyl groups and showed the Ar-OH bond can be readily broken (in Fig. 4), it would

be more comprehensive if the dehydroxylation is included.

Response: We thank the reviewer for the comment very much. As mentioned by the reviewer, phenolic hydroxyl groups exist in the lignin structure. In addition, the cleavage of the aliphatic C-O bonds during the reaction of lignin can also form the phenolic hydroxyl groups. As a demonstration of the usability of the RuW/HY₃₀ catalyst in Fig. 4, the phenolic C_{sp2}-O bond can be cleaved along with the C_{sp2}-O(CH₃) bond, which should be attributed to the RuW catalyzed hydrogenolysis of the phenolic hydroxyl group using the active hydrogen derived from the formaldehyde molecules generated during the SSH reaction of the C_{sp2}-O(CH₃) bond (Supplementary Fig. 2). To explore the mechanism of dehydroxylation, we have performed the reaction of 4-(1-hydroxypropyl)phenol over the RuW/HY₃₀ catalyst in the formaldehyde solution, and the results have been listed in Supplementary Table 8. During the reaction, thermolabile paraformaldehyde was used as the source of formaldehyde in the solution to simulate the gradual supply of the formaldehyde generated from the SSH reaction. As expected, only the C_{sp2}-C_{sp3} bond in 4-(1-hydroxypropyl)phenol was deconstructed over the HY₃₀ component with phenol as the sole product, but the cleavage of the phenolic C_{sp2}-O bond did not occur without the source of formaldehyde (Supplementary Table 8, entry 1). With the introduction of formaldehyde, benzene product was detected (Supplementary Table 8, entry 2), suggesting that the phenol C_{sp2}-O bond could be hydrogenolyzed over the RuW component using the active hydrogen derived from formaldehyde (Angew. Chem. Int. Ed. 2015, 54, 9057-9060). Furthermore, the yield of the benzene product was gradually increased with the increase of the source of formaldehyde (Supplementary Table 8, entries 2-8), which confirmed that the phenolic C_{sp2}-O bonds can be efficiently hydrogenolyzed over the RuW component with the active hydrogen derived from the gradually increased formaldehyde molecules during the RuW catalyzed SSH reaction of the C_{sp2}-O(CH₃) bonds.

In the revised manuscript, according to the comment, we have discussed the dehydroxylation by “Meanwhile, the phenolic C_{sp2}-O bonds with higher bond energy could also be cleaved along with the C_{sp2}-O(CH₃) bond, which should be attributed to the RuW catalyzed hydrogenolysis of the phenolic hydroxyl group using the active hydrogen derived from the formaldehyde molecules generated during the SSH reaction of the C_{sp2}-O(CH₃) bond (Supplementary Fig. 2). To explore the mechanism of dehydroxylation, we conducted the reaction of 4-(1-hydroxypropyl)phenol over the RuW/HY₃₀ catalyst in the formaldehyde solution (Supplementary Table 8), where thermolabile paraformaldehyde was used as the source of formaldehyde to simulate the gradual supply of the formaldehyde generated from the SSH reaction. As expected, only the C_{sp2}-C_{sp3} bond in 4-(1-hydroxypropyl)phenol was deconstructed over the HY₃₀ component with phenol as the sole product, and the cleavage of the phenolic C_{sp2}-O bond did not occur without the source of formaldehyde (Supplementary Table 8, entry 1). With the introduction of formaldehyde, the benzene product was detected (Supplementary Table 8, entry 2), suggesting that the phenolic C_{sp2}-O bond could be hydrogenolyzed over the RuW component using the active hydrogen derived from formaldehyde. Moreover, the yield of benzene was steadily increased with the increase of the source of formaldehyde (Supplementary Table 8, entries 2-8), which confirmed

that the phenolic C_{sp2}-O bonds can be efficiently hydrogenolyzed over the RuW component with the active hydrogen derived from the gradually increased formaldehyde molecules during the RuW catalyzed SSH reaction of the C_{sp2}-O(CH₃) bonds.” (Please see Page 13, marked in red)

3. In situ refining

When the hydrogen gas is introduced, the selectivity of benzene is lowered and yields of other saturated compounds and alkylbenzene increased. There seem to be competing reactions between in-situ hydroprocessing and hydrogenolysis by external hydrogen gas.

Also, the SiO₂/Al₂O₃ ratio did affect the yield and selectivity of benzene without the external hydrogen gas, there are multiple factors that determines the reaction and product distributions.

Please discuss this more comprehensively regarding this matter.

Different reaction conditions were used to test several model compounds (Fig. 4). Why? What rationale? With the real lignin, more severe reaction conditions were used.

Response: We thank the reviewer for the comment very much. The comment contains two sub-comments, and we respond to them respectively in the followings.

1) When the hydrogen gas is introduced, the selectivity of benzene is lowered and yields of other saturated compounds and alkylbenzene increased. There seem to be competing reactions between in-situ hydroprocessing and hydrogenolysis by external hydrogen gas. Also, the SiO₂/Al₂O₃ ratio did affect the yield and selectivity of benzene without the external hydrogen gas, there are multiple factors that determines the reaction and product distributions. Please discuss this more comprehensively regarding this matter.

Response: We thank the reviewer for the comment very much. In the cases of traditional hydroprocessing processes with exogenous hydrogen introduced, hydrogenolysis of the C_{sp2}-O bonds usually competes with the inevitable hydrogenation of the benzene rings and hydrogenolysis of the hydroxyl group in [C_{sp2}-C_{sp3}(OH)] motif, which consequently leads to the saturated cycloalkane and alkylbenzenes (Table 1, entries 5 and 6). In this work, the in situ refining strategy could avoid the above competitive reactions and exclusively afford the benzene product, owing to the innovative cooperation of HY₃₀ zeolite and RuW NPs on the well-ordered deconstruction of the C_{sp2}-C_{sp3} and C_{sp2}-O bonds, where RuW NPs catalyzed SSH reaction of the C_{sp2}-O bonds with the in situ abstracted hydrogen from the reactant not only can completely avoid the saturation of the benzene ring, but more importantly allow the C_{sp2}-C_{sp3} bonds to be deconstructed promptly over the HY₃₀ zeolite without the competition from the hydrogenolysis of the hydroxyl group in [C_{sp2}-C_{sp3}(OH)] motif. However, the deconstruction efficiencies of the HY zeolites on the C_{sp2}-C_{sp3} bonds were different with the zeolite framework Si/Al ratios. Compared with RuW/HY₃₀ catalyst, although the RuW/HY catalysts with lower zeolite framework Si/Al ratios (3, 5 and 15) possessed more acid sites, their strong Bronsted acid sites were in lower proportions (Supplementary Table 3) and actually distributed in the micropores with higher proportions (Supplementary Table 4), which heavily restricted the deconstruction

efficiency of the $C_{sp2}-C_{sp3}$ bonds on the Bronsted acid sites due to the inferior diffusibility of the microporous structures. In these cases, the poor deconstruction efficiency of the $C_{sp2}-C_{sp3}$ bonds offered RuW the opportunity to hydrogenolyze the hydroxyl group the $[C_{sp2}-C_{sp3}(OH)]$ motif using the active hydrogen derived from the formaldehyde molecules generated during the SSH reaction of the $C_{sp2}-O$ bonds (Supplementary Fig. 2), consequently leading to the n-propylbenzene byproduct (Table 1, entries 11 to 13).

In the revised manuscript, we have discussed this by “This is rationalized by the inevitable competitions of the hydrogenation of the benzene rings and hydrogenolysis of the hydroxyl group substituted at the aliphatic α -C (C_α) position (namely the C_{sp3} carbon in the $C_{sp2}-C_{sp3}$ bond, Fig. 1c) in the traditional hydroprocessing processes using exogenous hydrogen as reductant¹⁶, which have been successfully avoided via the HY₃₀ zeolite and RuW NPs accomplished refining strategy for the in situ transformation of the $C_{sp2}-C_{sp3}$ and $C_{sp2}-O$ bonds.” (Please see Page 6, marked in red)

“Compared with RuW/HY₃₀ catalyst, although the RuW/HY catalysts with lower zeolite framework Si/Al ratios (3, 5 and 15) possessed more acid sites, their strong Bronsted acid sites were in lower proportions (Supplementary Table 3) and actually distributed in the micropores with higher proportions (Supplementary Table 4), which heavily restricted the deconstruction efficiency of the $C_{sp2}-C_{sp3}$ bonds on the Bronsted acid sites due to the inferior diffusibility of the microporous structures, matching that only less than one-third of **1a** were transformed into benzene, and the main product was n-propylbenzene (**1e**) (Table 1, entries 11 to 13).” (Please see Page 7, marked in red)

“As the comparison of the utility of the RuW/HY₃₀ catalyst, it is the poor activities of the supports in the above catalysts (Table 1, entries 9-13 and 15-24) on the deconstruction of the $C_{sp2}-C_{sp3}$ bonds that offered RuW the opportunity to hydrogenolyze the hydroxyl group substituted at the aliphatic C_{sp3} position of the $[C_{sp2}-C_{sp3}(OH)]$ motif using the active hydrogen derived from the formaldehyde molecules generated during the SSH reaction of the $C_{sp2}-O$ bonds (Supplementary Fig. 2), consequently leading to the n-propylbenzene byproduct (Supplementary Table 5).” (Please see Page 7, marked in red)

2) Different reaction conditions were used to test several model compounds (Fig. 4). Why? What rationale? With the real lignin, more severe reaction conditions were used.

Response: We thank the reviewer for the comment very much. In order to design the catalytic system and simplify the mechanism study on transformation of the $C_{sp2}-C_{sp3}$ and $C_{sp2}-O$ bonds, in this work, we initially used 1-(4-methoxyphenyl)-1-propanol (**1a**) as the model compound, because of its lignin-mimetic phenylpropanol structure $[(CH_3O)Ph-C_{sp3}(OH)-]$ with only one methoxy group $[C_{sp2}-O(CH_3)]$ and one 1-hydroxypropyl group $[C_{sp2}-C_{sp3}(OH)]$ substituted on the benzene ring. However, the molecular structures of the monomeric and dimeric model compounds with the combinatorial groups of methoxyl, hydroxyl and isopropoxyl substituted on the benzene ring, are more complicated than that of **1a** (Fig. 4). Therefore, the desired transformation of these complex model compounds required different reaction conditions (for example, higher temperature). Similarly, the molecular weight and

structure of the real lignin are far higher and more complicated in comparison to those of the model compounds, which consequently caused the severe reaction conditions in the desired production of benzene from lignin. There are many cases regarding the comparable condition changes in the transformation of lignin and its model compounds. Please reference the previous studies, such as Liao, Y. H., Sels, B. F. et al. *Science* **367**, 1385-1390 (2020); Wang, X. Y. & Rinaldi, R. *Angew. Chem. Int. Ed.* **52**, 11499-11503 (2013); Zhang, C. F., Wang, F. et al. *ACS Catal.* **7**, 3419-3429 (2017); Zhao, C., Lercher, J. A. et al. *Angew. Chem. Int. Ed.* **48**, 3987-3990 (2009); Dong, L., Wang, Y. Q. et al. *Chem* **5**, 1-16 (2019); Zhao, C., Lercher, J. A. et al. *J. Catal.* **280**, 8-16 (2011); Gao, F., Hartwig, J. F. et al. *ACS Catal.* **6**, 7385-7392 (2016); Nichols, J. M., Ellman, J. A. et al. *J. Am. Chem. Soc.* **132**, 12554-12555 (2010); Zhang, J. G., Yan, N. et al. *Green Chem.* **16**, 2432-2437 (2014); Li, L. X., Wang, Y. Q. et al. *ACS Catal.* **10**, 15197-15206 (2020); Song, Q., Wang, F. & Xu, J. *Chem. Commun.* **48**, 7019-7021 (2012); Li, H. L. & Song, G. Y. *ACS Catal.* **9**, 4054-4064 (2019).

4. What is the main role of water?

Response: We thank the reviewer for the comment very much. In this work, water plays two roles during the reactions, including reaction medium and reactant. Serving as the reaction medium in the catalytic system, water can form hydrogen bonds with the O atom in anisole, which is beneficial for the SSH reaction of the C_{sp2}-O bond. Besides, water is the reactant in the formation of the phenolic hydroxyl group during the hydrolysis of the aliphatic carbon-oxygen (C_β-O, Fig. 1c) bonds.

In the revised manuscript, we have discussed the role of water by “As the reaction medium in the catalytic system, water also served as an ideal booster for the SSH reaction of the C_{sp2}-O bond by way of forming hydrogen bonds with the O atom in anisole⁴⁹, which was beneficial for the in situ refining strategy.” (Please see Page 5, marked in red)

“...accompanied by the HY₃₀ catalyzed hydrolysis of the aliphatic carbon-oxygen (C_β-O, Fig. 1c) bonds⁵⁸. At this stage, water was the only reactant besides **11a** in the formation of the phenolic hydroxyl (-OH) groups, and no active hydrogen was needed.” (Please see Page 13, marked in red)

5. Lignin-to-benzene

The results indicate that the yield of benzene is not proportional to the content of phenylpropanol structure. What would be the main determining factor?

Mass balance analysis is necessary. The authors only discuss the production of benzene. To evaluate the overall process efficiently, the mass balance should be presented.

Response: We thank the reviewer for the comment very much. The comment contains two sub-comments, and we respond to them respectively in the followings.

1) The results indicate that the yield of benzene is not proportional to the content of phenylpropanol structure. What would be the main determining factor?

Response: We thank the reviewer for the comment very much. It is well known that lignin is mainly an amorphous tridimensional polymer of three primary units: sinapyl

(3,5-dimethoxy-4-hydroxycinnamyl), coniferyl (3-methoxy-4-hydroxycinnamyl) and *p*-coumaryl (4-hydroxycinnamyl) alcohols, joined by ether and C-C linkages, which are also known as syringyl (S), guaiacyl (G) and *p*-hydroxylphenyl (H) units, respectively. All these monomeric building blocks contain a phenyl group and a propyl side chain, but differ in the number of methoxy groups substituted on the benzene ring, that is, S unit has two methoxy groups, G unit has one methoxy group, and H unit has none [Chem. Rev. **115**, 11559-11624 (2015); Science **344**, 709-719 (2014); Chem. Rev. **110**, 3552-3599 (2010); Chem. Rev. **118**, 614-678 (2018)]. Production of benzene product from lignin is substantially dependent on the abstraction of the benzene ring from the aforementioned S, G and H units, where the mass yield of benzene is different based on the structure of these units. Specifically, the mass yield of benzene abstracted from S units is sequentially lower than those from the equivalent G and H units. The content of the S, G and H units in lignin is related to plant taxonomy. For example, eucalyptus lignin contains more S units (Supplementary Figure 20b), but pine lignin has more G and H units (Supplementary Figure 15b), which is the main determining factor for the lower mass yield of benzene obtained from eucalyptus lignin, albeit with larger content of the phenylpropanol structures (Supplementary Table 9).

In the revised manuscript, we have discussed this by “Notably, the yields of benzene product were not always proportional to the content of phenylpropanol structure (Supplementary Table 9), which is related to the contents of the S, G and H units in lignin. Specifically, the mass yield of benzene abstracted from S units is sequentially lower than those from the equivalent G and H units. In contrast, pine lignin has more G and H units (Supplementary Figure 15b), but eucalyptus lignin contains more S units (Supplementary Figure 20b), which leads to a lower mass yield of benzene obtained from eucalyptus lignin, albeit with larger content of the phenylpropanol structures (Supplementary Table 9).” (Please see Page 15, marked in red)

2) Mass balance analysis is necessary. The authors only discuss the production of benzene. To evaluate the overall process efficiently, the mass balance should be presented.

Response: We thank the reviewer for the comment very much.

According to the comment, in the revised manuscript, we have provided the mass balance analysis of the lignin transformation by “After the reaction of lignin, the gas was released, passing through the ethyl acetate. Then, the reaction mixture in the reactor was transferred into a centrifuge tube. After that, the reactor was washed with the ethyl acetate used for the gas filtration, which was finally combined with the reaction mixture. By centrifugation, the solid was separated from the reaction mixture, and the yield of the detectable products in the ethyl acetate layer was determined by GC. The separated solid was successively washed with acetone, and the used catalyst was recovered. Then, the collected liquid was subjected to rotavap to remove acetone solvent, and the lignin residue was obtained. Finally, the lignin residue and recovered catalyst were freeze-dried under vacuum for 24 hours. The mass of the recovered catalyst was nearly the same as that of the catalyst initially loaded. The mass balance for the transformation of lignin was $92\pm 5\%$, which was calculated using Eq. (1).” (Please see Page 19, marked

in red)

6. TEA

Please consider including technoeconomic insights of the proposed process.

Response: We thank the reviewer for the comment very much. Techno-economic analysis (TEA) is very important for the evaluation of the potential of practical application. Based on our refining strategy, lignin was fed to the catalytic reactor only with water, and exclusively converted into benzene product, where the catalyst could be recovered and reused in the next reaction. Meanwhile, as the only liquid product, benzene could be quite easily separated from the system without complex procedures. The sufficiently recyclable and highly selective features of the above processes are beneficial to producing benzene economically. From the perspective of atomic economy, the active hydrogen atoms in the lignin molecule could also be utilized successfully along with the abstraction of the benzene rings from lignin under the in situ refining strategy. Moreover, the lignin residue obtained in the lignin conversion process can be collected and further valorized into high value-added fuels and chemicals. Given the above advantageous features, this lignin-to-benzene route has the potential of industrial application. However, we cannot provide an accurate TEA analysis at present on the basis of the laboratory-scale experiments. We have considered the comment very carefully, and we will further perform the pilot experiment to precisely analyze the technical economy of our lignin-to-benzene route. Nevertheless, if the reviewer has further suggestions, please let us know.

According to the comment, in the revised manuscript, we have discussed the TEA of our proposed strategy by "...lignin was fed to the catalytic reactor only with water, and exclusively converted into benzene product, where the catalyst could be recovered and reused. Meanwhile, as the only liquid product, benzene could be quite easily separated from the system without complex procedures. The sufficiently recyclable and highly selective features of the above processes are beneficial to producing benzene economically. From the perspective of atomic economy, the active hydrogen atoms in the lignin molecule could also be utilized successfully along with the abstraction of the benzene rings from lignin under the in situ refining strategy. Moreover, the lignin residue obtained in the lignin conversion process can be collected and further valorized into high value-added fuel products and chemicals. Given the above advantageous features, this lignin-to-benzene route has the potential of industrial application." (Please see Page 16, marked in red)

Reviewer #3 (Remarks to the Author):

For future biorefineries it is of great economical importance to obtain value added products from the lignin part of lignocellulosic biomass. One such product could be benzene, which is currently produced from fossil resources. Previous efforts in this area have not been successful in producing significant quantities of benzene from processing of either biomass directly or from isolated lignin. In a previous contribution by the authors (ref 31 of the paper), they showed that a similar catalyst (RuW/SiO₂) was able

to produce arenes from lignin. In this paper, it is reported that high-silica HY zeolite supported RuW alloy catalyst enables in situ refining of lignin to produce benzene in significant yields—a maximum of 18.8 % on lignin weight basis. For several lignin-type model compounds, the yield of benzene is almost 100 %. The experimental results are highly interesting and of great novelty and represent a significant step forward compared to previous papers in refining lignin to value added products. The authors show that the catalysis occurs by Bronsted acid catalyzed transformation of the C_{sp2}-C_{sp3} bonds on the local structure of lignin molecule and RuW catalyzed hydrogenolysis of the C_{sp2}-O bonds using hydrogen abstracted in-situ from the lignin molecule. Interestingly, the chemistry takes place using water as solvent. The reaction mechanism is elucidated in detail by a combination of control experiments and density functional theory calculations. The authors also (modestly) up-scale their experiments to produce 8.5 g of benzene from 50 g of lignin.

Response: We thank the reviewer for the positive comment very much.

1. In the proposed process, lignin is first extracted from the biomass by a solvent and then processed. It would be very relevant if the authors could comment on the economy of this two-step process. What should the benzene yield be for the process to be cost neutral?

Response: We thank the reviewer for the comment very much. In this work, the lignin-to-benzene route integrates two steps, including lignin extraction and catalytic valorization of lignin, which can not only preserve the native structure of lignin for better understanding of the genuine reactivity of lignin, but more importantly, can free the lignin transformation, involving conversion process and product separation, from the interference of the reaction of the carbohydrate in wood powder. In the first step, lignin was extracted from wood powder by solid-liquid separation and solvent recuperation, during which the used wood powder was also recovered along with the organic solvent and then used in the continual extraction process. In the second step, the extracted lignin was fed to the catalytic reactor only with water, and exclusively converted into benzene product, where the catalyst could be recovered and reused. Meanwhile, as the only liquid product, benzene could be quite easily separated from the system without complex procedures. The sufficiently recyclable and highly selective features of the above processes are beneficial to producing benzene economically. From the perspective of atomic economy, the active hydrogen atoms in the lignin molecule could also be utilized successfully along with the abstraction of the benzene rings from lignin under the in situ refining strategy. Moreover, the lignin residue obtained in the lignin conversion process can be collected and further valorized into high value-added fuels and chemicals. Given the above advantageous features, this lignin-to-benzene route has potential of industrial application. However, we cannot provide an accurate economic analysis at present on the basis of the laboratory-scale experiments. We have considered the comment very carefully, and we will further perform the pilot experiment to precisely calculate the cost of our lignin-to-benzene route. Then, the accurate benzene yield will be calculated for the process to be cost neutral. However, if the reviewer has further suggestions, please let us know.

According to the comment, in the revised manuscript, we have discussed this by “Based on the experimental results, we know that the lignin-to-benzene route integrates two steps, including lignin extraction and catalytic valorization of lignin, which can not only preserve the native structure of lignin for better understanding of the genuine reactivity of lignin, but more importantly, can free the lignin transformation from the interference of the reaction of the carbohydrate in wood powder. In the first step, lignin was extracted from wood powder by solid-liquid separation and solvent recuperation, during which the used wood powder was also recovered along with the organic solvent and then used in the continual extraction process. In the second step, the extracted lignin was fed to the catalytic reactor only with water, and exclusively converted into benzene product, where the catalyst could be recovered and reused. Meanwhile, as the only liquid product, benzene could be quite easily separated from the system without complex procedures. The sufficiently recyclable and highly selective features of the above processes are beneficial to producing benzene economically. From the perspective of atomic economy, the active hydrogen atoms in the lignin molecule could also be utilized successfully along with the abstraction of the benzene rings from lignin under the in situ refining strategy. Moreover, the lignin residue obtained in the lignin conversion process can be collected and further valorized into high value-added fuel products and chemicals. Given the above advantageous features, this lignin-to-benzene route has the potential of industrial application.” (Please see Pages 15 and 16, marked in red)

2. Please state the mass of catalyst applied in the experiments in table 1. It seems it is also not stated in the extended data file. This is necessary.

Response: We thank the reviewer for the comment very much.

According to the comment, we have stated the mass of catalyst applied in the experiments in Table 1. (Please see the footnotes in Table 1 in Page 5, marked in red)

Besides, we also have rechecked the statements of the reaction conditions in other Tables (Supplementary Information). The masses of all the catalysts employed in the reactions have been stated explicitly in the footnotes.

3. The authors use expensive Ru in their catalysts. This may likely impede the industrial implementation of the discovered chemistry. Did the authors test any non-noble metals as substitute for Ru?

Response: We thank the reviewer for the comment very much. In the optimization studies, we had tested the catalytic performances of the bimetallic MW/HY₃₀ catalysts (M=Ni, Co, Fe, Mo and Cu) with Ru element substituted by the non-noble metals. However, these bimetallic MW/HY₃₀ (M=Ni, Co, Fe, Mo and Cu) catalysts could only catalyze the C_{sp2}-C_{sp3} bond deconstruction without reaction of C_{sp2}-O bond, and consequently gave the anisole product (Supplementary Table 1, entries 1-5). This suggests that these non-noble bimetallic catalysts cannot abstract the hydrogen from the methoxy group and achieve the SSH reaction of C_{sp2}-O bond. When the exogenous hydrogen was introduced into the catalytic system, the C_{sp2}-O bond could be hydrogenolyzed over the MW/HY₃₀ catalysts (for example, NiW/HY₃₀, CoW/HY₃₀ and

FeW/HY₃₀), however, their selectivities to benzene were much lower than that over RuW/HY₃₀ catalyst (Supplementary Table 1, entries 6-10). Based on the control experiments (Table 1 and Supplementary Table 1), RuW/HY₃₀ was the best catalyst for the transformation of C_{sp2}-O and C_{sp2}-C_{sp3} bonds in lignin, which exclusively produced benzene in high yield. Although Ru is expensive, only a catalytic amount of Ru metal is used in our catalyst, and more importantly, the RuW/HY₃₀ catalyst can be recycled and reused in our catalytic system. Thus, from this point of view, the industrial potential of the employed RuW/HY₃₀ catalyst in this work deserves to be further studied and explored in the future.

According to the comment, we have provided the catalytic performances of these non-noble catalysts, and the results have been listed in Supplementary Table 1. In the revised manuscript, we have discussed this by “Likewise, the bimetallic MW/HY₃₀ (M=Ni, Co, Fe, Mo and Cu) catalysts (Supplementary Table 1, entries 1-5) also only catalyzed the C_{sp2}-C_{sp3} bond deconstruction without reaction of C_{sp2}-O bond, proving that the combination of Ru and W was necessary for the SSH reaction of the C_{sp2}-O bond. Although the C_{sp2}-O bond could be hydrogenolyzed in the presence of the exogenous hydrogen over the MW/HY₃₀ catalysts (for example, NiW/HY₃₀, CoW/HY₃₀ and FeW/HY₃₀), their selectivities of benzene were much lower than that over RuW/HY₃₀ catalyst (Supplementary Table 1, entries 6-10).” (Please see Page 6, marked in red)

4. Why was the experiments with lignin operated at 240 °C, when the model compound experiments were done at 180 °C?

Response: We thank the reviewer for the comment very much. In order to design the catalytic system and simplify the mechanism study on transformation of the C_{sp2}-C_{sp3} and C_{sp2}-O bonds, in this work, we initially used 1-(4-methoxyphenyl)-1-propanol (**1a**) as the model compound, because of its lignin-mimetic phenylpropanol structure [(CH₃O)Ph-C_{sp3}(OH)-] with only one methoxy group [C_{sp2}-O(CH₃)] and one 1-hydroxypropyl group [C_{sp2}-C_{sp3}(OH)] substituted on the benzene ring. However, the molecular weight and structure of the real lignin are much higher and more complicated in comparison to those of the **1a** compound, which consequently caused the severe reaction conditions in the desired transformation of lignin into benzene product. There are many cases regarding the comparable condition changes in the transformation of lignin and its model compounds. Please reference the previous studies, such as Liao, Y. H., Sels, B. F. et al. *Science* **367**, 1385-1390 (2020); Wang, X. Y. & Rinaldi, R. *Angew. Chem. Int. Ed.* **52**, 11499-11503 (2013); Zhang, C. F., Wang, F. et al. *ACS Catal.* **7**, 3419-3429 (2017); Zhao, C., Lercher, J. A. et al. *Angew. Chem. Int. Ed.* **48**, 3987-3990 (2009); Dong, L., Wang, Y. Q. et al. *Chem* **5**, 1-16 (2019); Zhao, C., Lercher, J. A. et al. *J. Catal.* **280**, 8-16 (2011); Gao, F., Hartwig, J. F. et al. *ACS Catal.* **6**, 7385-7392 (2016); Nichols, J. M., Ellman, J. A. et al. *J. Am. Chem. Soc.* **132**, 12554-12555 (2010); Zhang, J. G., Yan, N. et al. *Green Chem.* **16**, 2432-2437 (2014); Li, L. X., Wang, Y. Q. et al. *ACS Catal.* **10**, 15197-15206 (2020); Song, Q., Wang, F. & Xu, J. *Chem. Commun.* **48**, 7019-7021 (2012); Li, H. L. & Song, G. Y. *ACS Catal.* **9**, 4054-4064 (2019).

5. It would be useful if the authors could comment on the other products formed from the processing real lignin. Were any other useful products than benzene formed?

Response: We thank the reviewer for the comment very much. In addition to the benzene product, some other useful products, for example 2, 6-dimethoxyphenol, 2-methoxyphenol and phenol, can be obtained during the reaction of the real lignin (within 12 h, Supplementary Fig. 14). As the reaction proceeds, the above intermediate products can be further converted into benzene product under the function of the in situ refining strategy. After 12 h, the real lignin can be efficiently transformed with benzene as the only liquid product (Supplementary Figs. 13 and 25). With the abstraction of the benzene rings from molecular structure, the real lignin is finally converted into the lignin residual which can be further valorized into high value-added fuel products and chemicals.

According to the comment, we have discussed this by “In addition to the benzene product, some intermediate products, for example 2, 6-dimethoxyphenol, 2-methoxyphenol and phenol, could be detected during the reaction (Supplementary Fig. 14). As the reaction proceeded, the above intermediates were further transformed with benzene as the only liquid product (Supplementary Fig. 13), coinciding with the course of the reaction of **11a** (Supplementary Fig. 11).” (Please see Page 15, marked in red)

“Moreover, the lignin residue obtained in the lignin conversion process can be collected and further valorized into high value-added fuel products and chemicals.” (Please see Page 16, marked in red).

Overall the present work has high novelty and provides sufficient detail to allow reproduction with the possible exceptions noted above. I recommend publication.

Response: We thank the reviewer for the positive comment very much.

REVIEWERS' COMMENTS

Reviewer #2 (Remarks to the Author):

The authors have addressed all the questions and concerns raised by this reviewer. For future work, since the title of this work is "Sustainable" production of benzene from lignin, it would be much more interesting to see the "sustainability" of the proposed process. (e.g., real TEA works)

Reviewer #3 (Remarks to the Author):

I find that the authors have responded well to the issues raised by the reviewers and may be accepted for publication.

Reviewer #2 (Remarks to the Author):

The authors have addressed all the questions and concerns raised by this reviewer. For future work, since the title of this work is "Sustainable" production of benzene from lignin, it would be much more interesting to see the "sustainability" of the proposed process. (e.g., real TEA works)

Response: We thank the reviewer for the positive and valuable comment very much. Techno-economic analysis (TEA) is very important for the evaluation of the potential of practical application. As suggested by the referee, in the future we will try our best to develop a sustainable route for producing benzene on the basis of this work.

Reviewer #3 (Remarks to the Author):

I find that the authors have responded well to the issues raised by the reviewers and may be accepted for publication.

Response: We thank the reviewer for the positive comment very much.